# DETECT, DECIDE, UNLEARN: A TRANSFER-AWARE FRAMEWORK FOR CONTINUAL LEARNING

**Yiwen Wang[1], Diana Benavides-Prado[2] & Yun Sing Koh[1]**
[1]University of Auckland, Auckland, New Zealand
[2]Queen Mary University of London, London, England
`{yiwen.wang, y.koh}@auckland.ac.nz, d.benavidesprado@qmul.ac.uk`

## ABSTRACT

Continual learning (CL) aims to continuously learn new tasks from data streams. While most CL research focuses on mitigating catastrophic forgetting, memorizing outdated knowledge can cause negative transfer, where irrelevant prior knowledge interferes with new task learning and impairs adaptability. Inspired by how the human brain selectively unlearns unimportant information to prioritize learning and recall relevant knowledge, we explore the intuition that effective CL should preserve and selectively unlearn prior knowledge that hinders adaptation. We introduce DEtect, Decide, Unlearn in Continual lEarning (DEDUCE), a novel CL framework that dynamically detects negative transfer and mitigates it by a hybrid unlearning mechanism. Specifically, we investigate two complementary negative transfer detection strategies: transferability bound and gradient conflict analysis. Based on this detection, the model decides whether to activate a Local Unlearning Module (LUM) to filter outdated knowledge before learning a new task. Additionally, a Global Unlearning Module (GUM) periodically reclaims model capacity to enhance plasticity. Our experiments demonstrate that DEDUCE effectively mitigates task interference and improves overall accuracy with an average gain of up to 4.55% over state-of-the-art baselines.

## 1 INTRODUCTION

Continual Learning (CL) enables models to continuously learn from data streams, while mitigating catastrophic forgetting (CF) (McCloskey & Cohen, 1989) and enhancing knowledge transfer (Chen et al., 2019). To address CF, various methods have been proposed that focus on retaining previously learned knowledge (Schwarz et al., 2018; Buzzega et al., 2020; Ramesh & Chaudhari, 2021). Despite their effectiveness, these methods often overlook another critical challenge: memorizing outdated information can hinder adaptation and lead to task interference (Wang et al., 2025). Imagine a driver assistance system that must continuously adapt to dynamic environments. Retaining outdated information, such as past lighting conditions, can hinder learning new traffic patterns and interfere with recalling relevant prior knowledge. In CL, this phenomenon manifests as negative transfer, where conflicts between old and new tasks degrade both forward and backward transfer, leading to suboptimal performance on both new and old tasks. We argue that addressing this should not only focus on preserving knowledge, but also on selectively unlearning interfering knowledge.

At first glance, introducing unlearning into CL seems counterintuitive: if one of the goals is to reduce CF, why deliberately unlearn? The key lies in what is unlearned. By selectively unlearning interfering knowledge before it causes conflict, task interference is reduced, thereby enabling better adaptation to new tasks and mitigating interference-driven forgetting of old tasks, as illustrated in Fig. 1. This intuition draws from dual perspectives. Neuroscience shows that when old and new experiences conflict, the human brain actively forgets irrelevant information to mitigate interference and support knowledge transfer (Gravitz, 2019; Feldman & Zhang, 2020). From machine learning, unlearning has been demonstrated to remove the influence of specific data without degrading overall performance (Shibata et al., 2021), and even to discard domain-specific knowledge to improve generalization (Basak & Yin, 2024). Motivated by these, we argue that unlearning should not be viewed as contradictory to knowledge retention. Instead, we propose a new perspective for CL: *how can continual learners selectively unlearn interfering knowledge when negative transfer occurs, thereby*

*enhancing adaptability and reducing task interference?* This leads us to two guiding questions: (i) How to detect negative transfer? (ii) How to selectively unlearn interfering prior knowledge?

To answer these questions, we introduce DE-DUCE (DEtect, Decide, Unlearn in Continual lEarning), a transfer-aware CL framework that dynamically identifies negative transfer and mitigates it by selective unlearning. DE-DUCE uses two complementary strategies for detecting negative transfer: a transferability bound, which estimates the task transferability gap based on their data distributions (Ben-David et al., 2010); and gradient conflict analysis, which identifies interference during optimization (Yu et al., 2020). To selectively unlearn interfering knowledge, DEDUCE employs a hybrid unlearning mechanism that operates unlearning at the batch-level, named Local Unlearning Module (LUM), and at the network-level, named Global Unlearning Module (GUM). Specifically, upon detecting negative transfer, LUM is triggered to unlearn outdated prior knowledge before new learning. In

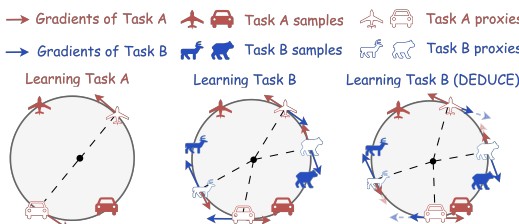

Figure 1: Illustration of negative transfer and the effect of unlearning. **Left**: Task A is learned, establishing its gradients and feature space. **Middle**: Directly learning Task B introduces gradient conflict with Task A, causing negative forward and backward transfer. **Right**: DEDUCE selectively unlearns interfering knowledge upon negative transfer detection, reducing the influence of old gradients on new learning and vice versa.

parallel, GUM enhances plasticity by monitoring neuron activity and resetting low-contributing and unimportant neurons, freeing capacity for future learning. Together, these modules enable DEDUCE to dynamically respond to negative transfer, promoting more efficient knowledge integration across tasks. Our main contributions are as follows:

- We introduce a transfer-aware CL framework that adaptively detects and responds to negative transfer. This framework provides a novel perspective for CL, where integrating selective unlearning into the learning process can improve both forward and backward transfer.
- We explore two complementary negative transfer detection strategies: transferability bound and gradient conflict analysis. These strategies provide CL with a systematic way to decide when to activate unlearning.
- We present a hybrid unlearning mechanism, which enables continual learners to dynamically regulate knowledge retention and unlearning, laying the foundation for a more adaptive continual learning system. This mechanism integrates LUM for batch-level interference removal and GUM for reclaiming underutilized capacity at the network level.

## 2 RELATED WORK

**Continual Learning.** Existing CL methods can be broadly divided into three categories: Memory-based (Rebuffi et al., 2017; Buzzega et al., 2020), Architecture-based (Mallya & Lazebnik, 2018; Kang et al., 2022), and Regularization-based methods (Zenke et al., 2017; Schwarz et al., 2018). Despite their effectiveness in mitigating CF, they are often limited in their ability to enhance knowledge transfer (Lopez-Paz & Ranzato, 2017) and adaptively handle interference from outdated information (Wang et al., 2025). To avoid negative transfer, Batch Spectral Shrinkage (Chen et al., 2019) shrinks dominant feature components; AFEC (Wang et al., 2021) prunes negatively contributing neurons based on gradient signals. For enhancing positive transfer, GPM (Saha et al., 2021) leverages subspace projections to mitigate interference; TRGP (Lin et al., 2022) introduces Trust Region Gradient Projection. However, these methods still prioritize maximizing knowledge reuse, without offering a flexible strategy that dynamically balances knowledge retention and adaptability.

**Machine Unlearning (MU).** MU was initially introduced to remove specific data traces for privacy compliance without full retraining (Guo et al., 2020; Wu et al., 2020). Beyond privacy (Bourtoule et al., 2021; Shibata et al., 2021), recent works have extended MU to broader scenarios, such as filtering domain-specific representations (Basak & Yin, 2024) and eliminating outdated knowledge (Wang et al., 2024c). These works demonstrate that unlearning prior knowledge can improve model performance. However, a critical limitation remains: most existing MU methods do not explicitly

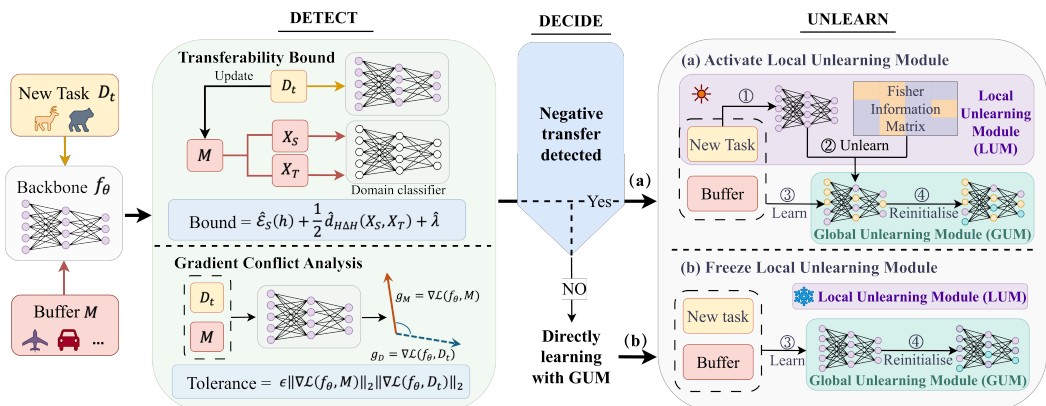

Figure 2: DEDUCE first detects potential negative transfer when a new task arrives, using either the transferability bound or gradient conflict analysis. If negative transfer is detected, the Local Unlearning Module (LUM) is activated, as shown in (a). The model first performs local unlearning in steps ①-②. Then, the new task is learned while randomly replaying previous tasks in step ③, with global unlearning (GUM) occurring throughout this process in step ④. If no negative transfer is detected, LUM remains frozen, as shown in (b), and the new task is learned directly with GUM.

consider when unlearning is necessary, which potentially limits positive transfer by discarding useful information excessively (Wang et al., 2021; 2024b). This highlights two key challenges that remain underexplored in CL: when to activate unlearning during the learning process, and how to leverage unlearning to enhance both forward and backward transfer. Motivated by these gaps, our work introduces a transfer-aware framework that detects potential negative transfer and dynamically regulates knowledge retention and unlearning to mitigate task interference and promote adaptability.

## 3 METHODOLOGY

To address negative transfer in CL, we design DEDUCE, a transfer-aware framework that integrates selective unlearning into the learning process. The core intuition is that, instead of indiscriminately preserving all prior knowledge, continual learners should dynamically unlearn interfering knowledge while retaining information that supports future learning. Implementing this intuition requires addressing two key challenges: (i) detecting negative transfer to identify conflicts between old and new tasks before they cause catastrophic interference, and (ii) deciding how to selectively unlearn outdated knowledge while preserving critical knowledge.

DEDUCE couples negative transfer detection with a hybrid unlearning mechanism. Detection is achieved via two complementary strategies. First, we introduce a transferability bound, inspired by domain adaptation theory (Ben-David et al., 2010). To make it practical, we employ LEEP (Nguyen et al., 2020) as a proxy, enabling task-level compatibility estimation in CL, which differs fundamentally from its original use in offline transfer analysis. Second, unlike GEM (Lopez-Paz & Ranzato, 2017), which constrains updates to rigidly preserve old tasks, we repurpose gradient conflict analysis to detect interference and trigger unlearning. This shifts the focus from strict preservation to adaptive regulation of stability and plasticity. Upon detecting negative transfer, the Local Unlearning Module (LUM) is activated to selectively unlearn batch-level interference and facilitate immediate adaptation. In parallel, the Global Unlearning Module (GUM) continuously monitors neuron contribution, reclaiming underutilized neurons to sustain long-term plasticity. Together, these modules allow DEDUCE to dynamically balance stability and adaptability across tasks, mitigating negative transfer while preserving essential knowledge. Fig. 2 provides an overview of the DEDUCE framework.

### 3.1 PRELIMINARIES

CL aims to learn from a dynamic data stream $\mathcal{D} = \{\mathcal{D}_1, \ldots, \mathcal{D}_t, \ldots, \mathcal{D}_T\}$, where each task dataset $\mathcal{D}_t = \{\boldsymbol{x}_t^i, y_t^i, t\}_{i=1}^{N_t}$ is sampled from a task-specific distribution $p_t(\boldsymbol{x}, y)$. Here, $\boldsymbol{x}_t^i \in \mathcal{X}$ is the $i$-th input of task $t$, $y_t^i \in \mathcal{Y}_t \subseteq \mathcal{Y} = \bigcup_{t=1}^{T} \mathcal{Y}_t$ is its corresponding label, and $t \in \mathcal{T} = \{1, 2, \ldots, T\}$ represents

the task identity. The objective is to learn a model $f_{\boldsymbol{\theta}}$, parameterized by $\boldsymbol{\theta}$, while the learner only has access to the current task data and an optional memory buffer $M$. In task-incremental learning (TIL), the task identity $t$ is available during both training and testing. A TIL system learns a function $f : \mathcal{X} \times \mathcal{T} \to \mathcal{Y}$, such that for a test instance, the model predicts $f(\boldsymbol{x}, t) \in \mathcal{Y}_t$. In class-incremental learning (CIL), the task identity is unavailable at test time. Thus, a CIL system must learn a unified predictor $f : \mathcal{X} \to \mathcal{Y}$ that distinguishes all previously seen classes.

## 3.2 DETECTING NEGATIVE TRANSFER TO DECIDE WHEN TO SELECTIVELY UNLEARN

Intuitively, when the new task introduces classes that are disjoint from those of previous tasks, updating the model can easily lead to negative transfer. This arises due to the task interference, which hinders the new knowledge learning and can even exacerbate CF (Chaudhry et al., 2019). Therefore, detecting negative transfer between new and prior tasks is a clear signal for selectively unlearning interfering prior knowledge. Based on the above analysis, CL can be naturally framed within the transfer learning paradigm, where previous tasks correspond to the source domain $X_{\mathcal{S}}$ and new task correspond to the target domain $X_{\mathcal{T}}$. Following the theoretical error bound (Ben-David et al., 2010), we can compute the transferability bound of target domain based on source domain. By comparing this bound with the target test error, we can assess whether the knowledge transfer from the prior tasks to the new task is negative. If the comparison suggests a high risk of negative transfer, selective unlearning is activated to mitigate interference before new learning. Specifically, for any hypothesis $h \in \mathcal{H}$, the theoretical target error $\mathcal{E}_{\mathcal{T}}(h)$ can be upper bounded by:

$$\mathcal{E}_{\mathcal{T}}(h) \leq \mathcal{E}_{\mathcal{S}}(h) + \frac{1}{2} d_{\mathcal{H}\Delta\mathcal{H}}(X_{\mathcal{S}}, X_{\mathcal{T}}) + \lambda \tag{1}$$

where $\mathcal{E}_{\mathcal{S}}(h)$ is the expected source error under hypothesis $h$, $d_{\mathcal{H}\Delta\mathcal{H}}$ denotes the $\mathcal{H}\Delta\mathcal{H}$-divergence, which measures the discrepancy between the source and target distributions, and $\lambda$ is the error of the ideal joint hypothesis $h^*$, defined as $\lambda = \mathcal{E}_{\mathcal{S}}(h^*) + \mathcal{E}_{\mathcal{T}}(h^*)$. In practice, directly computing $\lambda$ is intractable since it requires access to the optimal hypothesis over both source and target domains. To address this, we approximate the theoretical transferability bound by estimating three practical quantities (Ben-David et al., 2010): (1) the error of source domain $\hat{\mathcal{E}}_{\mathcal{S}}(h)$; (2) the divergence between the source and target domains $\hat{d}_{\mathcal{H}\Delta\mathcal{H}}(X_{\mathcal{S}}, X_{\mathcal{T}})$; (3) the transferability of the source model to the target $\hat{\lambda}$. By estimating these terms, we can approximate the bound and use it to predict the risk of negative transfer. We now explain how each component is computed in our framework. We estimate $\hat{\mathcal{E}}_{\mathcal{S}}(h)$ by evaluating the model's error on previous tasks. To calculate $\hat{d}_{\mathcal{H}\Delta\mathcal{H}}(X_{\mathcal{S}}, X_{\mathcal{T}})$, we train a domain classifier $h_d$ to discriminate between samples from the source domain $X_{\mathcal{S}}$ and samples from the target domain $X_{\mathcal{T}}$. The test error $\hat{\epsilon}(h_d)$ of this classifier can be used to approximate the divergence as (Theoretical proof in Appendix A.1):

$$\hat{d}_{\mathcal{H}\Delta\mathcal{H}}(X_{\mathcal{S}}, X_{\mathcal{T}}) = 2|1 - 2\hat{\epsilon}(h_d)| \tag{2}$$

where $\hat{\epsilon}(h_d)$ denotes the classification error of $h_d$ on a validation set composed of source and target instances. Intuitively, $d_{\mathcal{H}\Delta\mathcal{H}}$ measures how well a classifier can distinguish whether an input comes from the source or target domain. A high divergence indicates a large distribution shift and a high risk of negative transfer. To estimate $\hat{\lambda}$, we employ LEEP (Nguyen et al., 2020).

**Definition 1** (LEEP Score) For a given source model $f_{\boldsymbol{\theta}_S}$, the LEEP score measures the transferability between the source model and the target data $X_{\mathcal{T}}$, which is given by:

$$E(f_{\boldsymbol{\theta}_S}, X_{\mathcal{T}}) = \frac{1}{n} \sum_{i=1}^{n} \log \sum_{y_s \in \mathcal{Y}_S} P(y^i | y_s) P(y_s | \boldsymbol{x}^i) \tag{3}$$

where $P(y^i | y_s)$ is the estimated probability of target label $y^i \in \mathcal{Y}_{\mathcal{T}}$ given the source label $y_s \in \mathcal{Y}_S$, $P(y_s | \boldsymbol{x}^i)$ is the model's predicted probability distribution over source classes for input $\boldsymbol{x}^i \in X_{\mathcal{T}}$, and $n$ is the number of target instances. From its definition, the LEEP score is always negative and a larger value (i.e., smaller absolute value) indicates better transferability. We can approximate $\hat{\lambda}$ as a function of LEEP using an empirical relationship:

$$\hat{\lambda} \approx c|E(f_{\boldsymbol{\theta}_S}, X_{\mathcal{T}})| \tag{4}$$

where $|E(f_{\boldsymbol{\theta}_S}, X_{\mathcal{T}})|$ indicates how well the label distribution predicted by the source model aligns with the target domain, and $c$ is a scaling factor. Combining the source error $\hat{\mathcal{E}}_{\mathcal{S}}(h)$, the estimated

divergence $\hat{d}_{\mathcal{H}\Delta\mathcal{H}}(X_S, X_T)$, and the estimated $\hat{\lambda}$, we obtain a practical proxy for the theoretical transferability bound of the target error as:

$$
\begin{aligned}
\mathcal{E}_T(h) &\leq \hat{\mathcal{E}}_S(h) + \frac{1}{2}\hat{d}_{\mathcal{H}\Delta\mathcal{H}}(X_S, X_T) + \hat{\lambda} \\
&\approx \hat{\mathcal{E}}_S(h) + |1 - 2\hat{\epsilon}(h_d)| + c|E(f_{\theta_S}, X_T)|.
\end{aligned}
\tag{5}
$$

### 3.2.1 DETECTING NEGATIVE TRANSFER ONLINE

According to the transferability bound, if the target test error exceeds this bound, it signals the presence of potential negative transfer (Zhang et al., 2023). In such cases, DEDUCE activates the LUM to proactively unlearn interfering prior knowledge. However, accurately estimating the target error requires access to the full target data, typically achieved by one full pass (epoch) of the task data, which limits its use in strict online CL settings, where each sample must be processed only once and learning decisions must be made instantly (Wang et al., 2024a).

To complement this and to enable real-time detection of potential negative transfer, we introduce a complementary strategy based on gradient conflict analysis. By comparing gradients from the current mini-batch with those from previous tasks, DEDUCE can dynamically detect negative transfer, allowing timely adaptation without access to the complete target domain. Intuitively, when the new task involves classes that are disjoint from previously learned ones, the gradient induced by the new task will often conflict with the gradient of previous tasks. Updating the model with such conflicting gradients can cause negative forward and backward transfer. Therefore, detecting the presence of gradient conflict before fully training on the new task offers a practical signal: when negative gradient conflicts are observed, it suggests that direct learning of the new task may harm prior knowledge and lead to poor adaptation; thus, selective unlearning should be triggered. Based on the memory buffer $M$, we define the loss computed over stored exemplars from previous tasks as follows:

$$
\mathcal{L}(f_\theta, M) = \frac{1}{|M|}\sum_{(\boldsymbol{x}, y) \in M} \mathcal{L}(f_\theta(\boldsymbol{x}), y).
\tag{6}
$$

While minimizing the loss on the new task, we need to avoid negative gradient conflict between the new task and previous tasks. To formally characterize this conflict between tasks, we introduce the following task gradients condition inspired by the gradient constrains of the update model (Lopez-Paz & Ranzato, 2017).

**Definition 2** (Negative Transferability) For any new task $\mathcal{D}_t = \{\boldsymbol{x}_t^i, y_t^i\}_{i=1}^{N_t}, t \in [1, T]$, we say it has negative conflict with previous tasks $M$ if for some $\epsilon \in [-1, 0]$

$$
\langle \nabla\mathcal{L}(f_\theta, M), \nabla\mathcal{L}(f_\theta, \mathcal{D}_t) \rangle \leq \epsilon \|\nabla\mathcal{L}(f_\theta, M)\|_2 \|\nabla\mathcal{L}(f_\theta, \mathcal{D}_t)\|_2.
\tag{7}
$$

Specifically, Definition 2 introduces negative transferability between tasks, in the sense that learning the task $t$ is said to be at risk of negative transfer if the model gradient of the new task $t$ conflicts with the model gradient of previous tasks. Formally, when the tolerance $\epsilon = 0$, if the correlation between the gradient of the new task and the gradients of previous tasks is negative, it indicates strong dissimilarity or opposition in the update directions. In this case, continuing to optimize the model on the new task will likely lead to negative forward and backward transfer.

It is worth noting that the transferability bound calculated by Eq.(5) and the negative transferability evaluated by Eq.(7) can provide two different strategies for detecting negative transfer before learning the new task $t$, which can be chosen according to the specific task settings and demands. Further analysis of these two strategies for detecting negative transfer is presented in Appendix A.4.

### 3.3 HOW TO SELECTIVELY UNLEARN?

To address the challenge of selectively unlearning outdated and interfering knowledge while preserving critical information, we propose a hybrid unlearning mechanism that enhances knowledge retention and adaptation in CL. This mechanism consists of two complementary modules: local unlearning and global unlearning. Upon detecting negative transfer, the Local Unlearning Module (LUM) is activated to unlearn outdated knowledge before learning a new batch, thereby mitigating batch-level interference and enabling rapid adaptation. In parallel, the Global Unlearning Module (GUM) continuously monitors neuron contribution and periodically reinitializes inactive and

unimportant neurons, reclaiming underutilized capacity and restoring plasticity. Together, LUM and GUM enable the model to balance knowledge preservation with plasticity, thereby facilitating more effective integration of new tasks while reducing interference from outdated information.

### 3.3.1 LOCAL UNLEARNING MODULE (LUM)

When negative transfer occurs, directly fine-tuning on a new task can be hindered by interference from prior knowledge, leading to poor adaptation and CF. To address this, LUM is activated to proactively unlearn interfering knowledge before new task learning. From a theoretical perspective, the unlearning process minimizes the KL divergence between the current CL model parameter posterior $\rho_t(\boldsymbol{\theta})$ and the target unlearned model parameter posterior $\rho_u(\boldsymbol{\theta})$. Following Wibisono (2018), we represent the target unlearned posterior as an energy-based form $\rho_u(\boldsymbol{\theta}) = e^{-\omega}$, where $\omega = -L_{CL}$. This KL divergence can be further decomposed as:

$$\begin{aligned} \mathrm{KL}(\rho_t \| \rho_u) &= \int \rho_t(\boldsymbol{\theta}) \log \frac{\rho_t(\boldsymbol{\theta})}{\rho_u(\boldsymbol{\theta})} d\boldsymbol{\theta} \\ &= -\int \rho_t(\boldsymbol{\theta}) \log \rho_u(\boldsymbol{\theta}) d\boldsymbol{\theta} + \int \rho_t(\boldsymbol{\theta}) \log \rho_t(\boldsymbol{\theta}) d\boldsymbol{\theta} \\ &= -E_{\rho_t} \log \rho_u + E_{\rho_t} \log \rho_t \\ &= -E_{\rho_t} L_{CL} + E_{\rho_t} \log \rho_t \end{aligned} \quad (8)$$

which corresponds to optimizing an energy functional that increases the loss on the current mini-batch, thereby encouraging the model to step toward the target unlearned parameter distribution. Inspired by this KL-based formulation, the unlearning loss is defined as:

$$\mathcal{L}_{unlearn} = -\mathcal{L}_{CE}(f_{\boldsymbol{\theta}_t}(\boldsymbol{x}_t), y_t) + \alpha D_{\boldsymbol{\Phi}}\left(\boldsymbol{\theta}_t, \boldsymbol{\theta}_t^k\right) \quad (9)$$

where $D_{\boldsymbol{\Phi}}\left(\boldsymbol{\theta}_t, \boldsymbol{\theta}_t^k\right) = \left\| \boldsymbol{\theta}_t - \boldsymbol{\theta}_t^k \right\|_2^2$, $\alpha \geq 0$ represents a form of regularization to prevent the model from unlearning the early mini-batches knowledge of the current task, and $\boldsymbol{\theta}_t^k$ refers to the model parameters after learning the first $k$ mini-batches. In CL, interfering knowledge refers to previously acquired information whose parameter gradients conflict with those of the new task, thereby hindering its learning and inducing negative transfer (Riemer et al., 2019). In contrast, useful knowledge denotes information encoded in parameters that are highly sensitive and important to previous tasks, i.e., parameters with high Fisher Information Matrix (FIM) values (Kirkpatrick et al., 2017). To ensure selective unlearning, our LUM is explicitly regularized using the FIM. Then, we take the gradient with respect to $\boldsymbol{\theta}_t$ for the RHS of the Eq.(9), we can obtain the following unlearning for the previously learned tasks:

$$\boldsymbol{\theta}_t' = \boldsymbol{\theta}_t + \delta F^{-1}[\alpha \nabla_{\boldsymbol{\theta}_t} D_{\boldsymbol{\Phi}}\left(\boldsymbol{\theta}_t, \boldsymbol{\theta}_t^k\right) - \nabla_{\boldsymbol{\theta}_t} \mathcal{L}_{CE}\left(f_{\boldsymbol{\theta}_t}(\boldsymbol{x}_t), y_t\right)] \quad (10)$$

where $F$ is the diagonal approximation of the FIM. In Eq.(10), we leverage the FIM and unlearning rate $\delta$ to regulate the unlearning process so as to progressively unlearn interfering prior knowledge before learning the new task. Therefore, the unlearning update focuses precisely on low-importance, interfering prior knowledge that would cause negative transfer, while high-FIM parameters that encode useful information are protected. Following local unlearning on the current mini-batch, the model is trained on this batch by minimizing the following objective function:

$$\mathcal{L}_{learn} = \mathcal{L}_{CE}(f_{\boldsymbol{\theta}_t'}(\boldsymbol{x}_t), y_t) + \beta(\boldsymbol{\theta}_t' - \boldsymbol{\theta}_{t-1})^T F(\boldsymbol{\theta}_t' - \boldsymbol{\theta}_{t-1}) \quad (11)$$

where $\mathcal{L}_{reg} = (\boldsymbol{\theta}_t' - \boldsymbol{\theta}_{t-1})^T F(\boldsymbol{\theta}_t' - \boldsymbol{\theta}_{t-1})$ is a regularization term used to encourage the use of parameters that are not important to previous tasks to learn new knowledge, $\beta \geq 0$, and $\boldsymbol{\theta}_{t-1}$ refers to the optimal model parameters that were learned for the previous $t-1$ tasks.

### 3.3.2 GLOBAL UNLEARNING MODULE (GUM)

As models learn sequentially, their plasticity naturally decreases, making adaptation to new information increasingly difficult. This limitation is further compounded by task interference and finite network capacity. To address this, we propose monitoring neuron activity to identify neurons that remain largely inactive across recent tasks, based on the intuition that such neurons contribute little to learning the current task (Dohare et al., 2024). However, reinitializing neurons solely based on low-activation risks eliminates those that are sparsely active yet critical for specific tasks.

To mitigate this, we introduce importance scores that more accurately assess each neuron's contribution before reinitialization. Activity scores measure the actual influence of neurons on model outputs during the current task, while importance scores quantify their historical significance. Neurons identified as both low-activation and unimportant are selectively reinitialized, reclaiming model capacity for future learning without interfering with essential prior knowledge. As it continuously monitors neuronal contributions throughout training and globally reinitializes low-contributing neurons, we refer to this module as Global Unlearning Module (GUM). To measure the contribution of each neuron during training, we maintain a running average updated with a decay rate $\eta$. Specifically, in a feed-forward network, the contribution $C_{l,i}^{\tau}$ of the $i$-th features in layer $l$ at time $\tau$ is updated as:

$$C_{l,i}^{\tau} = \left[ (1-\eta)|\boldsymbol{h}_{l,i}^{\tau}| \sum_{k=1}^{n_{l+1}} |\boldsymbol{w}_{l,i,k}^{\tau}| + \eta C_{l,i}^{\tau-1} \right] \sigma(\tilde{F}_{l,i}^{\tau}) \qquad (12)$$

where $\boldsymbol{h}_{l,i}^{\tau}$ denotes the output of the $i$-th feature in layer $l$ at time $\tau$, and $\boldsymbol{w}_{l,i,k}^{\tau}$ is the weight connecting the $i$-th neuron to the $k$-th neuron in layer $l+1$. The term $n_{l+1}$ is the number of neurons in layer $l+1$. The gating factor $\sigma(\tilde{F}_{l,i}^{\tau})$ is a sigmoid function of the normalized importance score $\tilde{F}_{l,i}^{\tau}$, which ensures that important neurons are less affected. The importance of the $i$-th neuron $F_{l,i}^{\tau}$ is calculated by aggregating the importance scores of all outgoing connections $F_{l,i}^{\tau} = \sum_{k=1}^{n_{l+1}} F_{l,i,k}^{\tau}$. To reclaim model capacity, we periodically reinitialize low-contributing neurons. When a neuron is reinitialized, its outgoing weights are reset to zero, effectively making it inactive. However, immediately after reinitialization, a neuron would naturally exhibit zero contribution, risking rapid repeated resets. To prevent this, we introduce a maturity threshold $m$: neurons are protected from further reinitialization until they have survived for at least $m$ update steps and are considered mature. At each update step, a fraction of mature neurons with less contribution $\phi$, called the global unlearning rate, is reinitialized in every layer. The Pseudocode of DEDUCE is shown in Appendix A.2.

## 4 EXPERIMENTS

**Baselines.** We compare DEDUCE with oEWC (Schwarz et al., 2018), A-GEM (Chaudhry et al., 2019), ER (Rolnick et al., 2019), DER++ (Buzzega et al., 2020), HAL (Chaudhry et al., 2021), PCR (Lin et al., 2023), OnPro (Wei et al., 2023), MOSE (Yan et al., 2024), and STAR (Eskandar et al., 2025). In addition, we report Joint Training (upper bound) and Fine-tuning (lower bound) as standard reference baselines.

**Evaluation Metrics.** To evaluate the learning performance, we use Averaged Accuracy (ACC) to measure the final accuracy across all tasks. Backward Transfer (BWT) measures the influence of learning the new task on all previous tasks (Lopez-Paz & Ranzato, 2017). To assess plasticity, we use the accuracy of new task learning as a proxy, since it reflects the model's ability to effectively acquire novel knowledge (Dohare et al., 2024).

**Datasets.** We evaluate the performance of DEDUCE on four datasets under TIL and CIL settings. Specifically, CIFAR-100 (Krizhevsky, 2009) is split into 10 tasks with 10 disjoint classes each; CIFAR-10 (Krizhevsky, 2009) into 5 tasks with 2 disjoint classes each; Tiny-ImageNet (Deng et al., 2009) into 10 tasks with 20 disjoint classes each; and CORE-50 (Lomonaco & Maltoni, 2017) into 24 tasks with overlapping classes across tasks. In addition, we validate DEDUCE on DN4IL (Gowda et al., 2023) under the domain incremental setting, as shown in Table 3. For the above datasets, we adopt ResNet-18, similar to Kang et al. (2022), and further evaluate DEDUCE with ViT-Base (Dosovitskiy et al., 2021), as shown in Table 4. More details are provided in Appendix A.3.1– A.3.4.

### 4.1 MAIN RESULTS

**Overall Performance.** Table 1 reports the final averaged accuracy (ACC) of all baseline methods across four datasets in the CIL and TIL settings. By integrating our proposed DEDUCE, we consistently enhance the overall ACC of memory-based (HAL, ER, DER++), regularization-based (oEWC, A-GEM, STAR), and online CL methods (PCR, MOSE, DER++) in the CIL and TIL settings. In particular, DEDUCE delivers substantial improvements to HAL, boosting accuracy by over 13.2% in the challenging CIL settings and over 8.3% in the TIL settings on both CIFAR-100 and CIFAR-10. Similar benefits are observed for the regularization-based method oEWC in the online CL setting,

Table 1: ACC on different datasets. All methods use a replay buffer of size 500. OUR(B) indicates DEDUCE using transferability bound for detecting negative transfer, OUR(G) indicates DEDUCE using gradient conflict analysis for detecting negative transfer, '—' indicates not applicable, and † indicates another setting in Appendix A.3.3. The first 11 rows correspond to evaluations under the online CL setting, while the remaining rows report results under the offline CL setting.

| Method | CIFAR-100 | | CIFAR-10 | | Tiny-ImageNet | | CORE-50 | |
|---|---|---|---|---|---|---|---|---|
| | CIL | TIL | CIL | TIL | CIL | TIL | CIL | TIL |
| Fine-tuning | 4.2±0.2 | 26.4±0.7 | 15.0±0.5 | 60.8±1.0 | 3.7±0.2 | 18.8±0.8 | 6.9±0.9 | 34.7±0.9 |
| PCR | 20.4±0.9 | — | 52.5±2.6 | — | 21.4±0.4 | — | 31.8±0.8 | — |
| w/OUR(G) | **21.0±0.4** | — | **53.6±1.0** | — | **21.8±1.2** | — | **32.3±0.6** | — |
| MOSE | 27.7±1.1 | — | 63.8±0.3 | — | 13.7±0.9 | — | 41.3±0.6 | — |
| w/OUR(G) | **29.3±0.6** | — | **64.7±0.1** | — | **14.4±0.5** | — | **43.2±0.4** | — |
| OnPro | 19.5±0.8 | — | 66.9±1.2 | — | 6.9±0.2 | — | 32.1±1.4 | — |
| w/OUR(G) | **21.4±0.1** | — | **68.4±0.8** | — | **7.6±0.2** | — | **34.7±0.5** | — |
| oEWC | 4.8±0.2 | 21.4±1.1 | 15.8±1.4 | 57.3±1.3 | 4.0±0.4 | 13.6±1.7 | 7.8±0.4 | 34.3±1.7 |
| w/OUR(G) | **5.6±0.6** | **27.9±1.3** | **16.4±1.0** | **62.9±1.1** | **4.8±0.5** | **20.3±1.4** | **7.9±0.3** | **35.0±1.3** |
| DER++ | 12.4±0.4 | 56.6±0.3 | 44.8±1.6 | 82.0±0.2 | 6.9±0.6 | 37.7±0.4 | 28.2±1.0 | 66.5±0.9 |
| w/OUR(G) | **15.1±0.2** | **57.3±0.3** | **50.5±0.3** | **86.8±0.6** | **8.8±0.4** | **41.9±0.4** | **29.8±0.8** | **70.7±0.9** |
| Fine-tuning | 9.3±0.1 | 37.5±1.5 | 19.6±0.0 | 61.2±2.8 | 8.1±0.0 | 18.9±0.9 | 8.7±0.1 | 33.4±0.3 |
| Joint training | 70.6±0.6 | 91.1±0.1 | 92.1±0.1 | 98.3±0.1 | 59.5±0.3 | 81.8±0.1 | 70.9±0.3 | 91.2±0.1 |
| A-GEM | 9.3±0.1 | 56.2±0.2 | 19.9±0.4 | 87.2±0.3 | 8.1±0.0 | 22.6±0.5 | 8.9±0.3 | 30.1±2.2 |
| w/OUR(G) | **9.5±0.1** | **59.3±0.4** | **21.4±0.2** | **88.0±0.7** | **8.4±0.2** | **25.1±0.5** | **9.1±0.1** | **55.4±0.4** |
| STAR(DER) | 38.8±0.7 | 76.4±0.3 | 73.1±0.8 | 94.2±0.9 | 16.7±0.9 | 53.3±0.8 | 36.8±0.7 | 73.3±0.3 |
| w/OUR(G) | **40.9±0.8** | **77.7±0.3** | **74.7±0.6** | **95.5±0.5** | **18.5±0.6** | **55.1±0.8** | **42.2±0.2** | **78.5±0.5** |
| STAR(XDER) | 41.4±0.6 | 84.8±0.3 | **63.8±1.6** | **95.4±0.3** | 19.7±0.4 | 55.7±0.3 | 38.4±0.6 | 75.9±0.4 |
| w/OUR(G) | **44.2±0.4** | **85.5±0.4** | 62.6±1.7 | 95.3±0.6 | **20.1±0.6** | **56.0±0.8** | **39.8±0.7** | **76.6±0.5** |
| HAL | 11.6±0.8 | 45.1±2.0 | 44.9±1.0 | 84.9±3.2 | 3.7±0.1 | 23.6±1.9 | 15.7±3.5 | 51.7±3.6 |
| w/OUR(B) | 15.8±0.1 | 61.3±0.2 | 53.2±1.3 | 90.4±0.1 | 6.7±0.1 | 34.8±0.9 | 16.0±2.2[†] | 51.7±3.4[†] |
| w/OUR(G) | **24.8±0.7** | **72.8±0.4** | **67.0±0.8** | **93.2±0.9** | **9.6±0.1** | **48.9±0.4** | **16.9±2.3** | **53.1±3.3** |
| ER | 21.4±0.3 | 73.4±0.5 | 57.7±0.3 | 93.6±0.3 | 9.8±0.4 | 50.0±0.4 | 29.9±0.7 | 71.3±0.6 |
| w/OUR(B) | 24.2±1.1 | 74.7±0.9 | 67.6±2.0 | **94.4±0.4** | 10.6±0.2 | 51.7±0.6 | **34.1±0.3**[†] | **74.9±0.8**[†] |
| w/OUR(G) | **26.1±0.5** | **76.0±0.2** | **70.4±0.4** | 94.1±0.6 | **10.6±0.0** | **51.8±0.6** | 33.4±0.7 | 73.9±0.9 |
| DER++ | 36.8±0.9 | 75.6±0.5 | 71.1±1.3 | 93.4±0.2 | 15.6±1.9 | 51.1±0.7 | 40.1±1.5 | 76.0±2.0 |
| w/OUR(B) | 37.8±0.7 | 75.8±0.1 | 74.3±0.0 | 94.4±0.2 | 19.7±0.8 | 53.5±0.6 | **43.9±1.0**[†] | **78.0±0.9**[†] |
| w/OUR(G) | **39.8±0.9** | **78.4±0.4** | **74.6±0.7** | **94.8±0.2** | **22.1±0.5** | **55.6±0.1** | 41.2±1.4 | 76.8±1.1 |

though the magnitude varies across datasets. Even when compared with the strong rehearsal-based baseline DER++, DEDUCE achieves margins of 3.0% and 3.5% in the CIL on CIFAR-100 and CIFAR-10, respectively. On the more challenging Tiny-ImageNet, which contains more classes per task and demands higher discriminative power, DEDUCE achieves a gain of 6.5% in the CIL and a gain of 4.5% in the TIL, showcasing its robustness under increased task complexity. In the case of CORE-50, where task boundaries are more ambiguous due to overlapping classes across tasks, DE-DUCE still delivers consistent improvements. The bound-based variant (OUR(B)) utilizes the transferability bound to detect potential negative transfer at the task level. In contrast, the gradient-based variant (OUR(G)) performs detection at the batch level, enabling much finer-grained identification of task interference. This adaptive, batch-wise control allows the model to respond immediately to emerging conflicts, which explains why OUR(G) achieves higher performance. Nevertheless, it is important to note that both variants consistently outperform the baselines, demonstrating that DE-DUCE serves as a versatile and effective enhancement framework across diverse CL paradigms and dataset characteristics.

**Comparison on Learning Process.** For a more comprehensive comparison, we evaluate the learning process on CIFAR-100 and Tiny-ImageNet as shown in Fig. 3. First, we report the ACC of each new task, where DEDUCE consistently outperforms all baselines, with particularly substantial gains observed for HAL. This suggests that the hybrid unlearning mechanism effectively enhances forward transfer and improves the model's plasticity. Next, we track the accuracy on Task 1 over time, as Task 1 is the most vulnerable to forgetting, making it a strong indicator of a method's

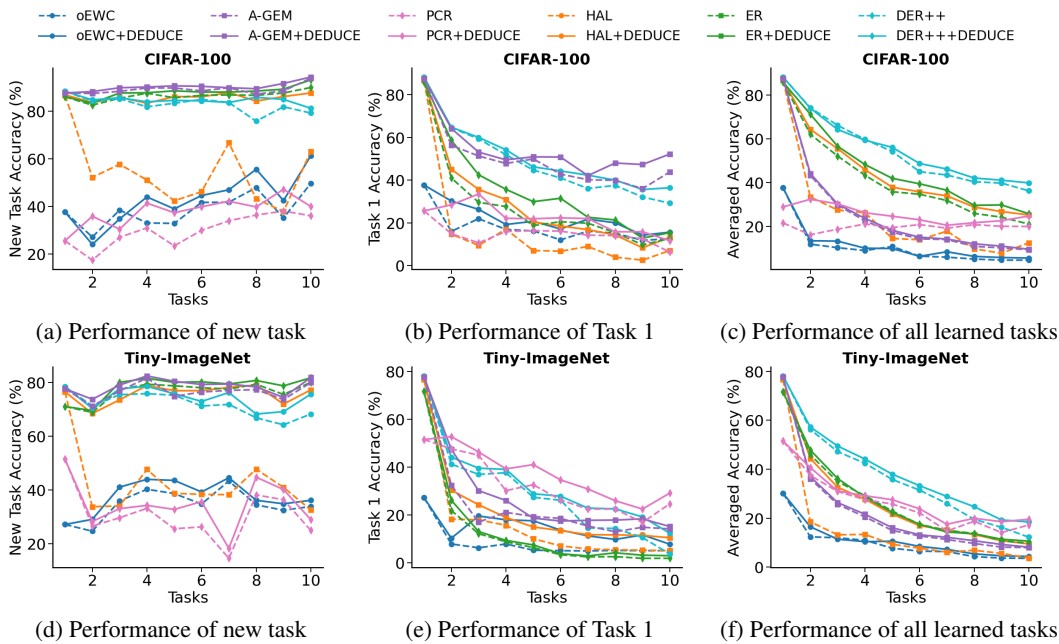

Figure 3: Performance on observed learning stages based on CIFAR-100 and Tiny-ImageNet.

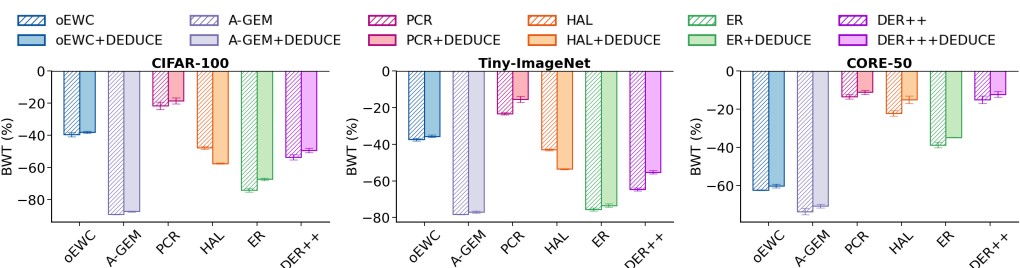

Figure 4: Backward transfer (BWT) on CIFAR-100, Tiny-ImageNet and CORE-50.

ability to mitigate CF. As illustrated in Fig. 3(b) and Fig. 3(e), although all methods suffer from forgetting, models equipped with DEDUCE exhibit a slower degradation. This indicates that DE-DUCE mitigates CF by reducing task interference through the timely activation of the LUM. Finally, as depicted in Fig. 3(c) and Fig. 3(f), ACC across all learned tasks shows that DEDUCE helps the model consistently maintain a more favorable balance between stability and plasticity over time.

**Comparison on Knowledge Transfer.** As shown in Fig. 3, DEDUCE enhances forward knowledge transfer when learning new tasks. To further evaluate the effect of DEDUCE, we report backward transfer (BWT) in Fig. 4. Across all datasets, baselines with DEDUCE consistently achieve higher BWT. For instance, DEDUCE improves DER++ by 6.4%, 9.2%, and 2.8% on CIFAR-100, Tiny-ImageNet, and CORE-50, respectively. These results demonstrate that DEDUCE not only avoids introducing additional CF but also helps mitigate negative backward transfer. Although HAL alone achieves slightly better BWT on CIFAR-100 and Tiny-ImageNet, its performance on forward trans-fer and overall accuracy remains inferior to that of HAL combined with DEDUCE (Detailed analysis is provided in Appendix A.3.5), suggesting that the overall benefits of integrating DEDUCE are still substantial.

## 4.2 ABLATION STUDIES

**Validation of DEDUCE.** We conduct a comprehensive ablation study to assess the contribution of each component in DEDUCE. Several key observations can be drawn from Table 2: (i) Each compo-

Table 2: Ablation study of the OUR proposed framework (DEDUCE).

| Method | CIFAR-100 | | CIFAR-10 | | Tiny-ImageNet | | CORE-50 | |
|---|---|---|---|---|---|---|---|---|
| | CIL | TIL | CIL | TIL | CIL | TIL | CIL | TIL |
| DER++ | 36.8±0.9 | 75.6±0.5 | 71.1±1.3 | 93.4±0.2 | 15.6±1.9 | 51.1±0.7 | 40.1±1.5 | 76.0±2.0 |
| $w$/LUM | 39.4±0.6 | 76.6±0.3 | 72.6±0.6 | 93.9±0.5 | 16.6±0.5 | 52.6±0.9 | 40.5±1.4 | 76.3±1.2 |
| $w$/GUM | 37.7±0.3 | 76.4±0.5 | 72.4±0.1 | 93.6±0.0 | 16.3±0.8 | 52.2±0.9 | **42.1±1.2** | **77.4±1.4** |
| $wo/\mathcal{L}_{reg}$ | 39.6±0.3 | 77.2±0.2 | 73.6±0.3 | 94.0±0.3 | 19.6±0.7 | 53.8±0.6 | 41.0±1.1 | 76.5±0.8 |
| $w$/OUR(G) | **39.8±0.9** | **78.4±0.4** | **74.6±0.7** | **94.8±0.2** | **22.1±0.5** | **55.6±0.1** | 41.2±1.4 | 76.8±1.1 |

nent of DEDUCE contributes positively to overall performance, with the combination of DER++ and DEDUCE yielding the most significant improvement. (ii) The LUM, which targets the unlearning of interfering prior knowledge, is especially beneficial in CIL, where the risk of task interference and negative transfer is more prominent. (iii) The GUM achieves consistent performance improvements across datasets, demonstrating the need to enhance the plasticity of the model while overcoming CF. More specific results are detailed in Tables 6–7 of Appendix A.3.5.

**Sensitivity Analysis.** We study the impact of three key hyperparameters: the local unlearning rate $\delta$, the global unlearning rate $\phi$, and the tolerance $\epsilon$ for triggering unlearning (Detailed in Tables 8–10 of Appendix). For both $\delta$ and $\phi$, performance initially improves and then declines as their values increase, suggesting that moderate unlearning effectively removes interference while preserving useful knowledge. For $\epsilon$, we observe that smaller values lead to performance degradation, as stricter activation of LUM reduces ACC. Notably, when $\epsilon = 0$, the model achieves the best ACC and BWT. These observations highlight the importance of investigating when to selectively unlearn, as excessive unlearning can harm performance, while timely unlearning can mitigate negative transfer.

**Impact of Buffer Size.** We report ACC and BWT of DEDUCE under varying memory capacities 100, 200, 500, 1000 (Detailed in Tables 11–12 of Appendix). Across all settings, DEDUCE consistently improves over baselines. Even with a large buffer size of 1000, where replay-based methods already perform strongly, DEDUCE delivers further gains, highlighting its complementary value. More importantly, under constrained memory of 100, DEDUCE achieves substantial improvements, effectively mitigating the degradation caused by limited replay. Overall, these results demonstrate that DEDUCE effectively mitigates task interference and facilitates knowledge integration, yielding robust continual learning across varying memory budgets.

## 5 CONCLUSION

In this work, we addressed the challenge of negative transfer in CL by introducing DEDUCE, a transfer-aware framework that proactively manages knowledge retention and update. DEDUCE detects negative transfer through transferability bound or gradient conflict analysis, and mitigates it with a hybrid unlearning mechanism that integrates dynamic local unlearning of interfering knowledge with periodic global unlearning of low-contributing knowledge. Unlike prior approaches that focus mainly on CF, DEDUCE emphasizes selective unlearning as a means to enhance both forward and backward knowledge transfer. Extensive experiments demonstrate that DEDUCE consistently improves state-of-the-art CL methods by reducing task interference. In future work, we plan to investigate mechanisms that explicitly promote beneficial transfer while suppressing interference, further advancing the adaptability of CL systems.

## ACKNOWLEDGEMENTS

This research was partially supported by the Program of China Scholarship Council (Grant No.202406050033). We also gratefully acknowledge the Advanced Machine Learning and Data Analytics Research (MARS) Lab at the University of Auckland.

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

# A   APPENDIX

## A.1   THEORETICAL PROOF

Following the theoretical error bound proposed by Ben-David et al. (2010), we aim to estimate the learning difficulty of the new task under the influence of prior knowledge. Specifically, for any hypothesis $h \in \mathcal{H}$, the theoretical target error $\mathcal{E}_\mathcal{T}(h)$ can be upper bounded by:

$$\mathcal{E}_\mathcal{T}(h) \leq \mathcal{E}_\mathcal{S}(h) + \frac{1}{2}d_{\mathcal{H}\Delta\mathcal{H}}(X_\mathcal{S}, X_\mathcal{T}) + \lambda. \tag{13}$$

where $\mathcal{E}_\mathcal{S}(h)$ is the expected source error under hypothesis $h$, $d_{\mathcal{H}\Delta\mathcal{H}}$ denotes the $\mathcal{H}\Delta\mathcal{H}$-divergence, which measures the discrepancy between the source and target distributions, and $\lambda$ is the error of the ideal joint hypothesis $h^*$, defined as $\lambda = \mathcal{E}_\mathcal{S}(h^*) + \mathcal{E}_\mathcal{T}(h^*)$.

To estimate $d_{\mathcal{H}\Delta\mathcal{H}}(X_\mathcal{S}, X_\mathcal{T})$, we train a domain classifier $h_d$ to discriminate between samples from the source domain $X_\mathcal{S}$ (labeled as 0) and samples from the target domain $X_\mathcal{T}$ (labeled as 1). The error $\hat{\epsilon}(h_d)$ of this classifier can be used to measure the divergence as:

$$\hat{d}_{\mathcal{H}\Delta\mathcal{H}}(X_\mathcal{S}, X_\mathcal{T}) = 2|1 - 2\hat{\epsilon}(h_d)|. \tag{14}$$

where $\hat{\epsilon}(h_d)$ denotes the classification error of $h_d$ on a set composed of source and target examples. Intuitively, $d_{\mathcal{H}\Delta\mathcal{H}}$ measures how well a classifier can distinguish between source and target domains. A high divergence indicates a large distribution shift and a high risk of negative transfer.

To show the proof of Eq.(14), we give the specific definitions of the symmetric difference hypothesis space $\mathcal{H}\Delta\mathcal{H}$ and $\mathcal{H}\Delta\mathcal{H}$-divergence (Kifer et al., 2004), which are useful in reasoning about error.

**Definition 3** ($\mathcal{H}\Delta\mathcal{H}$-divergence) Given a hypothesis space $\mathcal{H}$, the symmetric difference hypothesis space is defined as $\mathcal{H}\Delta\mathcal{H} = \{h(\boldsymbol{x}) \oplus h'(\boldsymbol{x}) \mid h, h' \in \mathcal{H}\}$, where $\oplus$ is the XOR operation. The $\mathcal{H}\Delta\mathcal{H}$-divergence between $X_\mathcal{S}$ and $X_\mathcal{T}$ is defined as

$$d_{\mathcal{H}\Delta\mathcal{H}}(X_\mathcal{S}, X_\mathcal{T}) = 2\sup_{h,h' \in \mathcal{H}} \left| \Pr_{\boldsymbol{x} \sim X_\mathcal{S}}(h(\boldsymbol{x}) \neq h'(\boldsymbol{x})) - \Pr_{\boldsymbol{x} \sim X_\mathcal{T}}(h(\boldsymbol{x}) \neq h'(\boldsymbol{x})) \right|. \tag{15}$$

As shown in Eq.(15), the definition of the $\mathcal{H}\Delta\mathcal{H}$-divergence measures the maximum difference in disagreement rates between the two distributions under the hypothesis class $\mathcal{H}$. By introducing the symmetric difference hypothesis class $\tilde{\mathcal{H}}$, where $\tilde{h}(\boldsymbol{x}) = 1[h(\boldsymbol{x}) \neq h'(\boldsymbol{x})], \tilde{h} \in \tilde{\mathcal{H}}$, the divergence can be rewritten as:

$$\begin{aligned} d_{\mathcal{H}\Delta\mathcal{H}}(X_\mathcal{S}, X_\mathcal{T}) &= 2\sup_{\tilde{h} \in \tilde{\mathcal{H}}} |\Pr_{\boldsymbol{x} \sim X_\mathcal{S}}(\tilde{h}(\boldsymbol{x}) = 1) - \Pr_{\boldsymbol{x} \sim X_\mathcal{T}}(\tilde{h}(\boldsymbol{x}) = 1)| \\ &= 2\sup_{A \in \mathcal{A}_{\tilde{\mathcal{H}}}} |\Pr_{\boldsymbol{x} \sim X_\mathcal{S}}(A) - \Pr_{\boldsymbol{x} \sim X_\mathcal{T}}(A)| \\ &= d_{\tilde{\mathcal{H}}}(X_\mathcal{S}, X_\mathcal{T}). \end{aligned} \tag{16}$$

where $\mathcal{A}_{\tilde{\mathcal{H}}} := \{\tilde{h}^{-1}(1)|\tilde{h} \in \tilde{\mathcal{H}}\}$ represents the set of the inputs that satisfy $\tilde{h}(\boldsymbol{x}) = 1$ induced by $\tilde{\mathcal{H}}$. This transformation expresses the divergence in terms of the probability of the symmetric difference classifier $\tilde{h}$ assigning a sample to the positive class (the set of $\tilde{h}(\boldsymbol{x}) = 1$) under the source and target distributions. The next lemma shows that we can compute the divergence by training a domain classifier which attempts to separate one domain from the other.

**Lemma 1** For a symmetric hypothesis class $\mathcal{H}$ and samples of size $n$ from $X_\mathcal{S}$, $X_\mathcal{T}$, the $\mathcal{H}$-divergence between $X_\mathcal{S}$ and $X_\mathcal{T}$ is

$$\hat{d}_{\mathcal{H}}(X_\mathcal{S}, X_\mathcal{T}) = 2\left(1 - \min_{h \in \mathcal{H}}\left[\frac{1}{n}\sum_{\boldsymbol{x} \sim h(\boldsymbol{x})=1} I[\boldsymbol{x} \in X_\mathcal{S}] + \frac{1}{n}\sum_{\boldsymbol{x} \sim h(\boldsymbol{x})=0} I[\boldsymbol{x} \in X_\mathcal{T}]\right]\right). \tag{17}$$

where $I[\boldsymbol{x} \in X_\mathcal{S}]$ is the binary indicator variable which is 0 when $\boldsymbol{x} \in X_\mathcal{S}$ and 1 when $\boldsymbol{x} \in X_\mathcal{T}$. Our basic plan of attack will be as follows: Label each source instance with 0 and the target instance with 1. Then train a domain classifier $h(\boldsymbol{x}) = \begin{cases} 0, & \boldsymbol{x} \in X_\mathcal{S} \\ 1, & \boldsymbol{x} \in X_\mathcal{T} \end{cases}$ under $h \in \mathcal{H}$ to discriminate between

source and target instances. The $\mathcal{H}\Delta\mathcal{H}$-divergence is immediately computable from the error as

$$\begin{aligned}
\hat{d}_{\mathcal{H}\Delta\mathcal{H}}(X_{\mathcal{S}}, X_{\mathcal{T}}) &= \hat{d}_{\tilde{\mathcal{H}}}(X_{\mathcal{S}}, X_{\mathcal{T}}) \\
&= 2|1 - (error_S + error_{\mathcal{T}})|.
\end{aligned} \tag{18}$$

where $error_S$ and $error_{\mathcal{T}}$ are the error rates of the domain classifier on the source and target domains, respectively. Since $\hat{\epsilon}(h_d)$ denotes the classification error of $h_d$ on a validation set of size $2n$ which is composed of source and target examples, we can obtain $\hat{\epsilon}(h_d) = \frac{error_S + error_{\mathcal{T}}}{2}$. Therefore, the $\mathcal{H}\Delta\mathcal{H}$-divergence can be calculated by Eq.(14).

## A.2 Pseudocode

---

**Algorithm 1** Local Unlearning Module (LUM)

---

**Require:** Sequence of tasks $\mathcal{D} = \{\mathcal{D}_1, \mathcal{D}_2, \ldots, \mathcal{D}_T\}$, Fisher Information Matrix $F$, local unlearning rate $\delta$, model learning rate $\rho$, model parameters $\boldsymbol{\theta}$
1: **for** $t = 1$ to $T$ **do**
2:      $X_t \leftarrow \text{LoadData}(\mathcal{D}_t)$
3:      **for** each $(\boldsymbol{x_t}, y_t) \in X_t$ **do**
4:          $\boldsymbol{\theta}'_t \leftarrow \boldsymbol{\theta}_t + \delta F^{-1}\nabla_{\boldsymbol{\theta}_t}\mathcal{L}_{unlearn}(\boldsymbol{\theta}_t)$
5:          $\boldsymbol{\theta}_t \leftarrow \boldsymbol{\theta}'_t - \rho\nabla_{\boldsymbol{\theta}'_t}\mathcal{L}_{learn}(\boldsymbol{\theta}'_t)$
6:      **end for**
7: **end for**
8: **return** $f_{\boldsymbol{\theta}_T}$

---

**Algorithm 2** Global Unlearning Module (GUM)

---

**Require:** Sequence of tasks $\mathcal{D} = \{\mathcal{D}_1, \mathcal{D}_2, \ldots, \mathcal{D}_T\}$, memory buffer $M$, global unlearning rate $\lambda$, decay rate $\eta$, Fisher Information Matrix $F$, maturity threshold $m$, network layers $L$, the number of neurons per layer $\{n_1, ..., n_L\}$, model parameters $\boldsymbol{\theta}$, model learning rate $\rho$
1: Initialize contributions $\{\boldsymbol{C}_1, ..., \boldsymbol{C}_L\}$, number of neurons to reinitialize $\{c_1, ..., c_L\}$, ages $\{a_1, ..., a_L\}$ to 0
2: Initialize the network weights $\{\boldsymbol{w}_1, ..., \boldsymbol{w}_L\}$, in which $\boldsymbol{w}_l$ is sampled from distribution $d_l$
3: **for** $t = 1$ to $T$ **do**
4:      $X_t \leftarrow \text{LoadData}(\mathcal{D}_t)$
5:      **for** each $(\boldsymbol{x}_t, y_t) \in X_t$ **do**
6:          $\boldsymbol{\theta}_t \leftarrow \boldsymbol{\theta}_t - \rho\nabla_{\boldsymbol{\theta}_t}\mathcal{L}_{learn}(\boldsymbol{\theta}_t)$
7:          update the weights $\{\boldsymbol{w}_1, ..., \boldsymbol{w}_L\}$ using stochastic gradient descent
8:          **for** $l = 1$ to $L$ **do**
9:              **for** $i = 1$ to $n_l$ **do**
10:                 Calculate importance score for neurons: $F_{l,i} = \sum_{k=1}^{n_{l+1}} F_{l,i,k}$
11:                 Normalized importance score for neurons: $\tilde{F}_{l,i} = \frac{F_{l,i} - minF_{l,i}}{maxF_{l,i} - minF_{l,i}}$
12:                 Update age: $a_{l,i} \leftarrow a_{l,i} + 1$
13:                 Update contribution by Equation (12)
14:              **end for**
15:              Find eligible neurons: $n_{eligible}$ = number of neurons with age more than $m$
16:              Update number of neurons to reinitialize $c_l \leftarrow c_l + \lambda n_{eligible}$
17:              **if** $c_l \text{¿} 1$ **then**
18:                 Find the neuron with smallest contribution and record its index as $\boldsymbol{r}$
19:                 Reinitialize input weights: resample $\boldsymbol{w}_{l-1}[\boldsymbol{r}]$ from distribution $d_l$
20:                 Reinitialize output weights: reset $\boldsymbol{w}_l[\boldsymbol{r}]$ to 0
21:                 Update number of neurons to reinitialize $c_l \leftarrow c_l - 1$
22:              **end if**
23:          **end for**
24:      **end for**
25: **end for**
26: **return** $f_{\boldsymbol{\theta}_T}$

---

---

**Algorithm 3** DEDUCE

---

**Require:** Sequence of tasks $\mathcal{D} = \{\mathcal{D}_1, \mathcal{D}_2, \ldots, \mathcal{D}_T\}$, memory buffer $M$, local unlearning rate $\delta$, global unlearning rate $\lambda$, maturity threshold $m$, Fisher Information Matrix $F$, model parameters $\boldsymbol{\theta}$, model $f_{\boldsymbol{\theta}}$

1: **for** $t = 1$ to $T$ **do**
2:      $X_t \leftarrow \text{LoadData}(\mathcal{D}_t)$
3:      **for** each $(\boldsymbol{x}_t, y_t) \in X_t$ **do**                          ▷ **Detect Phase**
4:          **if** UseTransferabilityBound **then**
5:              Calculate transferability bound $\mathcal{E}_{\mathcal{T}}(h)$ by Equation (5)
6:              $S \leftarrow \text{CompareBound}(\mathcal{E}_{\mathcal{T}}(h), \hat{\mathcal{E}}_{\mathcal{T}}(h))$
7:          **else if** UseGradientConflictAnalysis **then**
8:              $S \leftarrow \text{CompareGradient}(f_{\boldsymbol{\theta}}, D_t, M)$ by Equation (7)
9:          **end if**                                      ▷ **Decide Phase**
10:          **if** $S = $ there exists potential negative transfer **then**
11:              ActivateLUM $\leftarrow$ True
12:          **else**
13:              ActivateLUM $\leftarrow$ False
14:          **end if**                                     ▷ **Unlearn Phase**
15:          **if** ActivateLUM **then**
16:              $\boldsymbol{\theta}_t \leftarrow \text{LUM}(\boldsymbol{\theta}_t, F, \delta)$
17:          **end if**
18:          $\boldsymbol{\theta}_t \leftarrow \text{GUM}(\boldsymbol{\theta}_t, \lambda, m)$
19:          $M \leftarrow \text{UpdateBuffer}(M, \boldsymbol{x}_t, y_t)$
20:      **end for**
21:      Update Fisher Information Matrix
22: **end for**
23: **return** $f_{\boldsymbol{\theta}_T}$

---

## A.3    SETUPS AND ADDITIONAL EXPERIMENTS

### A.3.1    BASELINES

We compare DEDUCE with the following baselines on the datasets mentioned above. (1) oEWC (Schwarz et al., 2018): an online regularization-based method that estimates parameter importance by FIM. (2) A-GEM (Chaudhry et al., 2019): constrains the new task learning with the gradient calculated using the stored data of previous tasks. (3) ER (Rolnick et al., 2019): interleaves the data stored in the buffer with the current data during training. (4) DER++ (Buzzega et al., 2020): extends ER by also storing and distilling past model logits. (5) HAL (Chaudhry et al., 2021): aligns current representations with class-specific prototypes (anchors) from replayed data. (6) PCR (Lin et al., 2023): an online CL method that maintains class-level proxies and applies contrastive loss to mitigate CF in an online CIL setting. (7) OnPro (Wei et al., 2023): an online CL method that uses online prototypes to learn stable representations and maintain class discriminability. (8) MOSE (Yan et al., 2024): an online CL method that orchestrates latent expertise with multi-level supervision and reverse self-distillation. (9) STAR (Eskandar et al., 2025): stabilizes training by introducing stability-inducing weight perturbations.

### A.3.2    EVALUATION METRICS

To evaluate the learning performance, we use Averaged Accuracy (ACC) to measure the final accuracy averaged over all tasks. In addition, we use the Backward Transfer (BWT) to measure the average influence of learning the new task on all previous tasks. To assess plasticity, we use the accuracy of new task learning as a proxy, since it reflects the model's ability to effectively acquire novel knowledge. To evaluate forgetting, we track the accuracy on Task 1 over time, as Task 1 is the most vulnerable to forgetting, making it a strong indicator of a method's ability to mitigate CF.

$$\text{ACC} = \frac{1}{T} \sum_{i=1}^{T} A_{T,i} \tag{19}$$

$$\text{BWT} = \frac{1}{T-1} \sum_{i=1}^{T-1} (A_{T,i} - A_{i,i}) \tag{20}$$

where $A_{i,j}$ represents the testing accuracy of task $j$ after learning task $i$.

### A.3.3 DATASETS

We evaluate the performance of DEDUCE on CIFAR-100 (Krizhevsky, 2009), CIFAR-10 (Krizhevsky, 2009), Tiny-ImageNet (Deng et al., 2009), and CORE-50 (Lomonaco & Maltoni, 2017) in both TIL and CIL settings. Furthermore, we validate DEDUCE on DN4IL dataset (Gowda et al., 2023) under the domain incremental setting. The results demonstrate that DEDUCE still maintains its advantage in mitigating negative transfer under such challenging settings, as shown in Table 3.

Notably, CORE-50 is comprised of 50 classes, with around 2400 examples per class. For training, we split the dataset into 24 sequential tasks, where different tasks have joint classes. For testing, we split this dataset into 9 sequential tasks. Since different tasks have overlapping classes, when we use transferability to detect negative transfer, we cannot know which samples in the buffer come from the new task; thus, dividing the source and target domains becomes a great challenge. To evaluate the effectiveness of the transferability bound, we conduct experiments on CORE-50 in another setting. Assuming that classes sharing semantic similarity are more likely to induce positive knowledge transfer, we compute the transferability bound by classifying the new classes as the target domain and the previous classes that have been learned as the source domain when we detect new classes in a new task. Therefore, the results of DEDUCE(B) on CORE-50 in Table 1 are derived based on the above setup. We can see from Table 1 that DEDUCE(G) can deliver consistent improvement in the case of CORE-50, where the task boundaries are more ambiguous. Furthermore, DEDUCE(B) also shows notable improvements under the new setting, demonstrating that it effectively mitigate interference and promote forward transfer. This suggests that DEDUCE is robust in complex scenarios involving potential negative transfer.

Table 3: Validation of DEDUCE on domain incremental setting.

| Methods | ACC | BWT |
|---------|-----|-----|
| A-GEM | 26.98±0.82 | -37.96±0.67 |
| *w*/OUR(G) | **28.25±0.58** | **-33.21±0.19** |
| ER | 30.11±0.78 | -31.19±0.66 |
| *w*/OUR(G) | **32.12±0.33** | **-26.38±0.65** |
| DER++ | 40.03±0.53 | -20.02±0.44 |
| *w*/OUR(G) | **42.33±0.14** | **-16.68±0.37** |

### A.3.4 IMPLEMENTATION DETAILS

For all CL methods, we use the ResNet18 as backbone on all datasets, similar to Kang et al. (2022). Furthermore, we investigate the ViT-Base (Dosovitskiy et al., 2021) for CIFAR-100 and report the results in Table 4. We train the model with an SGD optimizer for all the datasets and compare methods on an Nvidia Tesla A100 GPU 40GB. For all the methods compared, we set the same batch size (32) and replay batch size (32) for fair comparisons. We reproduce all baselines in the same environment with their source code and adopt the hyperparameters from DER++ codebase (Buzzega et al., 2020) as the baseline settings for all the methods we compared in the experiments. For CIFAR-100 and CIFAR-10, we train the network for 50 epochs. For Tiny-ImageNet, we train the network for 100 epochs. For CORE-50, we train the network for 20 epochs. For DN4IL, we train the network for 50 epochs. In the experiments, we set buffer size $M = 500$, local unlearning rate $\delta = 0.001$, global unlearning rate $\phi = 10^{-5}$, and $\epsilon = 0.0$. We run each setting five times and report the average as well as the standard deviation.

Table 4: Performance of ViT-Base DEDUCE on CIFAR-100.

| Method | ACC | | BWT | |
|---|---|---|---|---|
| | CIL | TIL | CIL | TIL |
| DER++ | 9.87±0.87 | 43.03±0.23 | -51.56±0.33 | -16.77±0.27 |
| *w*/OUR(G) | **11.56±0.47** | **44.86±0.18** | **-50.14±0.18** | **-14.69±0.33** |

### A.3.5 MORE DETAILED EXPERIMENTAL RESULTS

**Comparison on Knowledge Transfer** As shown in Fig. 3, DEDUCE enhances forward knowledge transfer when learning new tasks. To further evaluate the effect of DEDUCE, we report backward transfer (BWT) in Fig. 4. Across all datasets, most baselines with DEDUCE consistently achieve higher BWT. However, when integrated with the proposed method, the BWT of HAL is decreased.

BWT measures the difference between a task's accuracy immediately after its training and its final accuracy after all tasks. Because our method substantially enhances HAL's learning of new tasks, this initial accuracy becomes much higher, increasing the BWT gap even when both the initial and final accuracies improve. For example, for Task 2 in CIFAR-100 (Table 5), HAL's accuracy drops from 52.2→2.9 (Δ=49.3), while HAL+OUR(G) drops from 83.6→13.2 (Δ=70.4). Although both 83.6 and 13.2 exceed the HAL baseline, the larger initial gain (83.6 vs. 52.2) yields a numerically larger drop, and thus a lower BWT.

Table 5: Performance of Task 2.

| Method | T1 | T2 | T3 | T4 | T5 | T6 | T7 | T8 | T9 | T10 |
|---|---|---|---|---|---|---|---|---|---|---|
| HAL | — | 52.2 | 15.4 | 10.5 | 3.7 | 7.2 | 6.6 | 4.0 | 2.1 | 2.9 |
| w/ OUR(G) | — | **83.6** | **45.1** | **30.9** | **20.4** | **18.5** | **16.8** | **14.3** | **8.3** | **13.2** |

This effect also relates to HAL's inherent design limitations. HAL anchors past activations to preserve old knowledge, providing strong stability but placing much tighter constraints on the learning of new tasks than on the retention of old ones. In other words, HAL's static anchoring restricts the model's ability to flexibly acquire new representations far more than it prevents forgetting. When our transfer-aware unlearning mechanism is integrated, it alleviates these constraints, leading to significant improvement in learning new tasks and moderate improvement in retaining old ones.

Therefore, although HAL achieves slightly better BWT on CIFAR-100 and Tiny-ImageNet, its performance on forward transfer and overall accuracy remains inferior to that of HAL+DEDUCE, as shown in Fig. 5, suggesting that the overall benefits of integrating DEDUCE are still substantial.

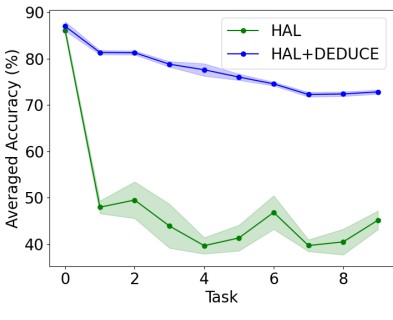
(a) Performance of all learned tasks

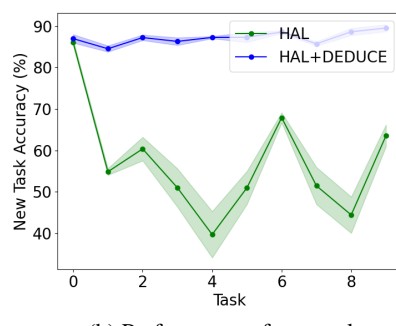
(b) Performance of new task

Figure 5: Performance of HAL on CIFAR-100.

**Ablation Study** We conduct a comprehensive ablation study to assess the contribution of each component in DEDUCE. Table 2 reports the overall accuracy (ACC), while Tables 6– 7 present the backward transfer (BWT) results in the CIL and TIL settings. As shown in Table 2, each mod-

ule of DEDUCE contributes positively to overall accuracy. The combination of DEDUCE with the strong baseline DER++ yields the most significant accuracy gains on most datasets. An exception is observed on CORE-50, where the combination of DER++ and the GUM achieves the best performance. We hypothesize that this is due to the unique characteristics of CORE-50, where task boundaries are ambiguous: many tasks contain overlapping or even identical classes. This overlap may undermine the effectiveness of the LUM because it will inadvertently unlearn the knowledge of the same classes as the current task. In contrast, the GUM is agnostic to task semantics and focuses on periodically reclaiming model capacity, which remains effective regardless of class overlap. Importantly, even in this challenging scenario, DEDUCE still outperforms the DER++ baseline, demonstrating its robustness in complex and overlapping task settings. Furthermore, BWT results in Tables 6– 7 confirm that DEDUCE consistently alleviates negative backward transfer across all datasets, suggesting that DEDUCE is effective in mitigating task interference.

Table 6: Ablation study of the OUR proposed framework (DEDUCE).

| Method | CIFAR-100 | | CIFAR-10 | |
|---|---|---|---|---|
| | CIL | TIL | CIL | TIL |
| DER++ | -53.71±1.60 | -14.72±0.96 | -25.55±1.37 | -4.89±0.43 |
| $w$/LUM | -49.76±0.87 | -14.21±0.24 | -25.29±0.91 | -4.50±0.33 |
| $w$/GUM | -52.45±1.18 | -14.54±0.54 | -24.79±0.10 | -4.32±0.21 |
| $wo$/$\mathcal{L}_{reg}$ | -47.61±0.93 | -14.11±0.62 | -21.31±0.53 | -3.21±0.25 |
| $w$/OUR(G) | **-47.36±1.43** | **-13.28±0.46** | **-20.84±0.59** | **-2.80±0.24** |

Table 7: Ablation study of the OUR proposed framework (DEDUCE).

| Method | Tiny-ImageNet | | CORE-50 | |
|---|---|---|---|---|
| | CIL | TIL | CIL | TIL |
| DER++ | -64.42±0.82 | -28.52±0.93 | -14.98±1.94 | -1.99±2.35 |
| $w$/LUM | -63.89±0.64 | -28.01±1.20 | -13.82±1.43 | -0.70±1.27 |
| $w$/GUM | -63.67±0.60 | -27.75±0.69 | **-11.97±1.32** | -1.14±1.20 |
| $wo$/$\mathcal{L}_{reg}$ | -58.28±0.88 | -26.57±0.78 | -12.35±1.10 | -0.47±1.38 |
| $w$/OUR(G) | **-55.21±0.98** | **-24.57±0.41** | -12.19±1.52 | **1.62±1.28** |

**Hyperparameter Analysis** We analyze the impact of three key hyperparameters: the local unlearning rate $\delta$ in Table 8, the global unlearning rate $\lambda$ in Table 9, and the tolerance $\epsilon$ for triggering local unlearning in Table 10. For both $\delta$ and $\lambda$, we can observe that the performance first increases and then declines as the values grow, indicating that moderate unlearning effectively removes interfering knowledge while preserving useful information. Regarding $\epsilon$, we observe that ACC and BWT degrade as $\epsilon$ decreases. Specifically, a smaller $\epsilon$ imposes stricter criterion for activating LUM, which can lead to performance degradation. In particular, when $\epsilon = 0$, the model achieves the highest ACC and BWT. This suggests that it is beneficial to trigger local unlearning whenever there exists potential negative transfer, even if the gradient conflict is not strong. It is worth noting that different baselines based on different datasets will have different optimal hyperparameters.

Table 8: Analysis of local unlearning rate $\delta$ on CIL.

| Dataset | $\delta$ | 0.0001 | 0.001 | 0.01 | 0.03 | 0.05 |
|---|---|---|---|---|---|---|
| CIFAR-100 | ACC | 37.77 | 39.84 | **44.08** | 40.87 | 31.04 |
| | BWT | -53.76 | -49.36 | -22.98 | -20.24 | **-16.80** |
| CIFAR-10 | ACC | 71.07 | 74.60 | **74.87** | 70.45 | 66.01 |
| | BWT | -28.55 | -20.84 | **-19.17** | -26.36 | -31.08 |
| Tiny-ImageNet | ACC | 15.40 | **22.13** | 17.68 | 14.72 | 13.57 |
| | BWT | -58.11 | **-55.21** | -56.95 | -59.93 | -60.27 |

Table 9: Analysis of global unlearning rate $\phi$ on CIL.

| Dataset | $\phi$ | 1e-6 | 1e-5 | 1e-4 | 5e-4 |
|---|---|---|---|---|---|
| CIFAR-100 | ACC | 39.12 | **39.84** | 37.93 | 31.26 |
| | BWT | -50.70 | **-49.36** | -52.26 | -59.48 |
| CIFAR-10 | ACC | 71.15 | **74.60** | 71.95 | 70.58 |
| | BWT | -27.51 | **-20.84** | -27.59 | -28.38 |
| Tiny-ImageNet | ACC | 18.60 | **22.10** | 18.47 | 17.30 |
| | BWT | -57.70 | **-55.21** | -58.51 | -59.78 |

Table 10: Analysis of tolerance $\epsilon$ on CIL.

| Dataset | $\epsilon$ | 0.2 | 0.0 | -0.2 | -0.5 | -0.7 |
|---|---|---|---|---|---|---|
| CIFAR-100 | ACC | 37.10 | **39.84** | 35.97 | 35.64 | 35.54 |
| | BWT | -53.54 | **-49.36** | -55.33 | -55.89 | -57.33 |
| CIFAR-10 | ACC | 70.03 | **74.60** | 71.67 | 71.33 | 69.47 |
| | BWT | -29.63 | **-20.84** | -27.38 | -28.68 | -31.16 |
| Tiny-ImageNet | ACC | 19.11 | **22.10** | 17.49 | 16.51 | 12.67 |
| | BWT | -58.05 | **-55.21** | -60.57 | -61.96 | -63.90 |

**Impact of Buffer Size** We report the performance of DEDUCE under varying memory capacities 100, 200, 500, 1000 in Tables 11– 12. Across all settings, DEDUCE consistently enhances baselines performance. Notably, even with a large buffer size as 1000, where replay-based baselines already achieve strong performance due to abundant memory, DEDUCE provides further gains, demonstrating its complementary value. More importantly, under constrained memory as 100, DEDUCE yields even greater relative improvements, effectively mitigating the performance drop caused by limited exemplars. These results highlight DEDUCE's ability to reduce task interference and enhance knowledge integration, enabling robust continual learning across memory budgets.

Table 11: Overall accuracy on CIFAR-100 with different memory sizes.

| Method | $M = 100$ | $M = 200$ | $M = 500$ | $M = 1000$ |
|---|---|---|---|---|
| ER | 58.23±1.49 | 66.53±1.17 | 73.43±0.54 | 78.71±0.96 |
| $w$/OUR(G) | **62.65±1.47** | **66.72±1.33** | **76.03±0.23** | **78.83±0.72** |
| DER++ | 56.38±0.35 | 67.04±0.36 | 75.64±0.53 | 79.93±0.56 |
| $w$/OUR(G) | **60.26±0.65** | **70.85±0.21** | **78.43±0.36** | **80.19±0.61** |

Table 12: Backward Transfer on CIFAR-100 with different memory sizes.

| Method | $M = 100$ | $M = 200$ | $M = 500$ | $M = 1000$ |
|---|---|---|---|---|
| ER | -33.44±0.66 | -23.99±0.53 | -15.82±0.62 | -11.80±0.88 |
| $w$/OUR(G) | **-27.89±1.01** | **-22.75±1.14** | **-13.92±0.39** | **-10.39±0.75** |
| DER++ | -36.04±0.53 | -25.10±0.47 | -14.72±0.96 | -10.67±0.44 |
| $w$/OUR(G) | **-31.70±0.54** | **-20.44±0.30** | **-13.24±0.25** | **-10.47±0.61** |

**Impact of Training Epochs** To assess the robustness of DEDUCE, we investigate the effect of different training epochs on performance, as increasing the number of epochs helps reduce optimization instability. We therefore further evaluate whether DEDUCE can continue to provide benefits under extended training. Table 13 reports results for DER++ and DER++ with DEDUCE under 50, 70, 90, 110, and 130 training epochs on CIFAR-100.

Two observations can be made: (i) Stable improvements across training lengths. DEDUCE consistently outperforms DER++ in both ACC and BWT, demonstrating that its benefits persist as training epochs increase. (ii) Efficiency advantage. DEDUCE at 50 epochs achieves higher accuracy than

Table 13: Analysis of training epochs.

|  | **Method** | 50 | 70 | 90 | 110 | 130 |
|---|---|---|---|---|---|---|
| ACC | DER++ | 36.85 | 35.92 | 36.55 | 37.86 | 38.86 |
|  | $w$/OUR(G) | **39.84** | **39.35** | **37.89** | **38.83** | **39.57** |
| BWT | DER++ | -53.71 | -55.32 | -54.65 | -54.96 | -52.50 |
|  | $w$/OUR(G) | **-47.36** | **-49.08** | **-50.84** | **-48.97** | **-49.30** |

DER++ trained for 130 epochs, showing that it can reduce training cost while improving performance. These observations confirm that longer training alone cannot resolve task interference, and DEDUCE remains necessary even when training epochs are increased.

**Computation Efficiency of DEDUCE** To quantify the computational overhead introduced by our method, we report training time per epoch on the CIFAR-100 dataset for various baselines, with and without our proposed modules in Table 14.

Table 14: Computation efficiency of the proposed method (CIFAR-100, one epoch in seconds)

| Method | Baseline | w/ OUR(B) | w/ OUR(G) |
|---|---|---|---|
| PCR | 11.19s | — | 57.38s |
| MOSE | 112.32s | — | 235.23s |
| oEWC | 6.94s | — | 61.30s |
| STAR | 23.12s | 37.46s | 85.06s |
| AGEM | 10.99s | 29.17s | 74.77s |
| HAL | 13.60s | 17.66s | 86.50s |
| ER | 11.23s | 16.49s | 68.69s |
| DER++ | 11.90s | 19.73s | 74.88s |

As shown in Table 14, while our detect–decide–unlearn framework introduces additional computation, the overhead remains moderate. Importantly, the bound-based variant OUR(B) adds minimal extra cost, its runtime is very close to the original baselines (e.g., HAL: 13.60s → 17.66s; DER++: 11.90s → 19.73s). This provides a highly efficient option for settings with limited computational budgets. The gradient-based variant (OUR(G)) incurs higher overhead because it performs batch-wise detection. However, its runtime is still within a reasonable range, and several baselines combined with OUR(G) (e.g., DER++ + OUR(G), STAR + OUR(G)) are still faster than MOSE alone, indicating that the cost is far from prohibitive.

Specifically, our method requires one extra forward pass for detection and one forward/backward pass for unlearning when unlearning is triggered. In many cases, unlearning is not activated, resulting in only a single additional forward pass. To further balance efficiency and performance, we provide multiple detection strategies with different granularities (task-level, epoch-level, and batch-level). These options act as built-in optimization strategies: users can choose the appropriate granularity depending on computational constraints, while all variants still provide consistent performance improvements (Appendix A.4.2).

## A.4 ANALYSIS OF NEGATIVE TRANSFER DETECTION STRATEGIES

To motivate our design, we provide two perspectives that illustrate how selective unlearning alleviates negative forward and backward transfer. Each perspective naturally leads to a corresponding detection strategy in DEDUCE.

### A.4.1 TRANSFERABILITY BOUND

From the feature space perspective, negative transfer arises when tasks induce overlapping representations, leading to blurred decision boundaries. As shown in Fig. 6(a), after learning Task A, introducing Task B can cause Task A's features to drift and overlap with Task B's, thereby degrading both tasks' performance. Selective unlearning helps reduce such overlap, stabilizing Task A's representation while improving Task B's adaptation, as shown in Fig. 6(b). Concretely, upon de-

tecting negative transfer, DEDUCE selectively unlearns outdated prior knowledge that drives task interference. This prevents features of earlier tasks from drifting into the representation space of new tasks, thereby reducing overlap and sharpening task boundaries. In contrast to indiscriminate unlearning, which discards all prior knowledge, selective unlearning preserves useful representations and removes only the interfering knowledge, striking a balance between stability and adaptability. Motivated by this, we adopt a transferability bound to quantify task compatibility.

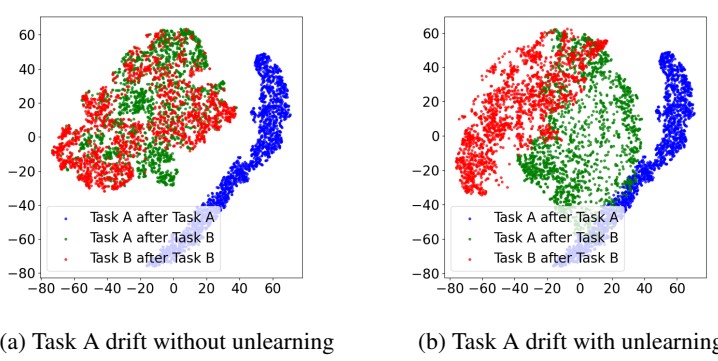

(a) Task A drift without unlearning      (b) Task A drift with unlearning

Figure 6: Toy example on feature-space effects of selective unlearning.

We further validate this strategy on CIFAR-100. The transferability bound predicts negative transfer when learning Task 2, but not Task 7. Consistent with this, Task 1 undergoes distribution drift and overlap with Task 2 without DEDUCE (Fig. 7(a)), while DEDUCE reduces the drift and overlap to some extent (Fig. 7(b)). In contrast, when learning Task 7, the bound does not trigger unlearning, and Task 6 remains stable with relatively clear separation from Task 7 (Fig. 7(c)). These results highlight two points: (i) The transferability bound provides a reliable signal for detecting negative transfer. (ii) DEDUCE can effectively reduce task interference and mitigate negative backward transfer.

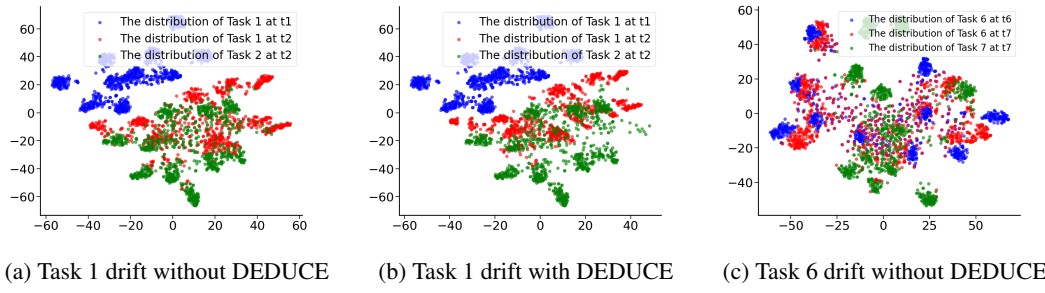

(a) Task 1 drift without DEDUCE    (b) Task 1 drift with DEDUCE    (c) Task 6 drift without DEDUCE

Figure 7: Analysis of transferability bound on CIFAR-100.

### A.4.2 GRADIENT CONFLICT ANALYSIS

From the optimization perspective, negative transfer manifests as conflicting gradients between old and new tasks. As illustrated in Fig. 1, when learning Task B after Task A, gradient directions may conflict, hinder adaptation, and induce forgetting. Selective unlearning mitigates this by unlearning interfering knowledge, thereby reducing gradient conflict and enabling smoother knowledge integration. This motivates our second detection strategy, gradient conflict analysis, which directly measures the alignment between task gradients. When strong conflicts are detected, the Local Unlearning Module (LUM) is activated to selectively unlearn outdated knowledge before learning the new batch, adaptively balancing stability and plasticity.

We evaluate two variants: epoch-level, which compares gradients once per epoch, and batch-level, which monitors conflicts at mini-batch granularity. Table 15 reports the frequency of LUM activation under each scenario, while Table 16 presents the corresponding accuracy achieved by DEDUCE. As

shown in Table 16, DEDUCE with both epoch-level and batch-level detection strategies can effectively improve accuracy across settings, validating the utility of gradient conflict as a proxy for task interference. Notably, the batch-level variant achieves higher accuracy improvements while activating the LUM less frequently, as shown in Table 15. This demonstrates its efficiency and precision in identifying negative transfer moments for unlearning. These results support the following conclusions: (i) Gradient conflict analysis provides a reliable online signal for detecting negative transfer. (ii) Timely activation of LUM is crucial, as immediate responses to detected interference enhance overall learning accuracy. (iii) DEDUCE can selectively unlearn interfering prior knowledge at the right time, thereby enabling better task adaptability and reducing interference.

Table 15: Frequency of LUM activation under different scenarios.

| Scenario | CIFAR-100 | CIFAR-10 | Tiny-ImageNet |
|---|---|---|---|
| Epoch-level | 92.22% | 84.66% | 47.97% |
| Batch-level | 35.99% | 26.55% | 36.68% |

Table 16: ACC of DEDUCE with gradient conflict analysis at epoch-level (G-E) and batch-level (G-B)

| Method | CIFAR-100 | | CIFAR-10 | | Tiny-ImageNet | |
|---|---|---|---|---|---|---|
| | CIL | TIL | CIL | TIL | CIL | TIL |
| DER++ | 36.85±0.98 | 75.64±0.56 | 71.14±1.36 | 93.46±0.29 | 15.61±1.92 | 51.10±0.74 |
| $w$/OUR(G-E) | 39.01±0.58 | 78.31±0.38 | 73.12±0.78 | 94.02±0.15 | 17.83±1.17 | 55.49±0.42 |
| $w$/OUR(G-B) | **39.84±0.93** | **78.44±0.63** | **74.56±0.73** | **94.77±0.24** | **22.09±0.46** | **55.60±0.05** |

### A.4.3 COMPUTATION EFFICIENCY OF DETECTION

We investigate two complementary strategies for detecting negative transfer. Table 17 compares the training time of the two detection strategies. The transferability bound is highly efficient because it operates at the task-level, providing a quick estimation of task transferability. In contrast, gradient conflict analysis, especially at batch-level, offers finer-grained and more stable detection, making it particularly suitable for online continual learning where task boundaries are unavailable. However, this comes at a higher computational cost, as reflected in the longer runtime. Overall, transferability bound is preferable when efficiency is critical, while gradient conflict analysis provides stronger precision and robustness in settings that demand accurate, online detection of interference.

Table 17: Computation efficiency of different detection strategies (CIFAR-100, one epoch).

| | DER++ | DEDUCE(B) | DEDUCE(G) |
|---|---|---|---|
| Time(seconds) | 11.90 | 14.32 | 52.95 |

### A.5 ETHICS STATEMENT

This work adheres to the ICLR Code of Ethics. It does not involve human subjects or sensitive data, and all datasets are publicly available. The proposed methods are intended for scientific purposes only, and we have ensured fairness, transparency, and compliance with ethical and legal standards.

### A.6 REPRODUCIBILITY STATEMENT

We present details of our experimental settings in Appendix A.3.3– A.3.4, as well as all the used hyper-parameters in Appendix A.3.5. Moreover, we present detailed pseudocode in Appendix A.2. Additionally, our source code is publicly available in the supplementary document.

### A.7 USE OF LARGE LANGUAGE MODELS

Large language models (LLMs) were used solely for language refinement and polishing of the manuscript. They did not contribute to the conception of the research, design of methods, execution of experiments, or analysis of results.

