# OpenReview forum: "Detect, Decide, Unlearn: A Transfer-Aware Framework for Continual Learning"
_ICLR.cc/2026/Conference — ICLR 2026 Poster_

### Official Review · Reviewer_wJFi · 2025-10-29

**Soundness:** 3
**Presentation:** 3
**Contribution:** 2
**Rating:** 6
**Confidence:** 4

**Summary:**

This paper introduces DEDUCE, a continual learning framework that detects negative transfer between tasks and mitigates it through selective unlearning. The core idea is that not all prior knowledge should be preserved - some interfering knowledge should be unlearned to improve adaptation to new tasks. The framework uses two detection strategies (transferability bounds and gradient conflict analysis) and two unlearning modules (local and global) to dynamically balance knowledge retention and plasticity. Experiments show consistent improvements across multiple datasets and baselines.

**Strengths:**

1. The paper addresses an underexplored aspect of continual learning - that blindly preserving all knowledge can hurt performance. This perspective is refreshing and the neuroscience motivation about selective forgetting is compelling.

2. The experimental evaluation is comprehensive, testing across multiple datasets (CIFAR-100, CIFAR-10, Tiny-ImageNet, CORE-50) and showing that DEDUCE works as a plug-in enhancement for various existing methods. The consistency of improvements across different baselines suggests the approach has broad applicability.

3. The paper provides two complementary detection strategies, which is useful given different computational and accuracy trade-offs. The ablation studies are thorough and help understand the contribution of different components.

**Weaknesses:**

1. The author may ignore the optimization-generalization gap. There's a fundamental conceptual issue with the gradient conflict detection mechanism. Gradient conflict is an optimization phenomenon, while negative transfer concerns generalization. The paper doesn't justify why opposing gradients during training would necessarily indicate poor test-time transfer between tasks. This conflation undermines the theoretical foundation of one of the main detection strategies.

2. The implementation of the transferability bound also has gaps. The use of LEEP scores as a proxy for λ isn't well justified, especially since LEEP was designed for offline transfer learning with fixed source models, not continual learning where models evolve. The connection between Eq.4 and the actual bound is hand-wavy at best.

3. The paper doesn't clearly explain what constitutes **interfering knowledge** versus **useful knowledge**. How does maximizing cross-entropy loss (Eq. 8) specifically target interference rather than just degrading performance broadly? The mechanism seems too coarse-grained for selective unlearning.

4. The global unlearning module appears disconnected from the negative transfer detection - it periodically resets neurons regardless of whether negative transfer is detected. This seems more like a generic plasticity mechanism than targeted unlearning.

5. The paper claims that unlearning improves plasticity but provides no direct evidence for this. While they show accuracy improvements on new tasks, they don't demonstrate that this is due to increased plasticity rather than other factors. Including analysis of network capacity in [1] could help.

[1] Dohare S, Hernandez-Garcia J F, Lan Q, et al. Loss of plasticity in deep continual learning[J]. Nature, 2024, 632(8026): 768-774.

**Questions:**

1. Can you provide theoretical or empirical evidence that gradient conflict during optimization actually correlates with negative transfer at test time?

2. How exactly does maximizing cross-entropy loss lead to selective unlearning of only interfering knowledge? What prevents this from removing useful knowledge as well?

3. In Tab. 13, the LUM activation frequency varies dramatically across datasets (92% for CIFAR-100 epoch-level vs 36% for batch-level). What does this suggest about the reliability of the detection mechanism?

4. Could the performance improvements be explained by regularization effects rather than actually addressing negative transfer or improving plasticity?

---

> ### Author Response · Authors · 2025-11-22
> **Official Comment by Authors**
>
> > W1 & Q1: The author may ignore the optimization-generalization gap. Can you provide theoretical or empirical evidence that gradient conflict during optimization actually correlates with negative transfer at test time?
>
> **Response:**
>
> We thank the reviewer for this important observation. Although optimization dynamics and generalization behavior are conceptually distinct, negative transfer in continual learning is fundamentally manifested through the optimization trajectory, and not only at test time.
>
> In continual learning, the generalization outcome on past and future tasks is directly shaped by how gradients from different tasks interact during training [1]-[2]. Therefore, optimization signals, particularly gradient conflict, provide actionable and reliable indicators of negative transfer [3]-[4].
>
> Specifically, knowledge transfer in continual learning occurs during training. When gradients from new and old tasks conflict, model parameters are updated in opposing directions, this overwrites prior representations, causing catastrophic forgetting of previous knowledge. At the same time, conflicting gradients dampen effective updates toward the new task’s objective, hindering the acquisition of new knowledge and leading to negative transfer.
>
> Our framework addresses this by detecting such conflicts before learning new tasks. When the optimization directions between past and incoming tasks are misaligned, this indicates a potential for both forward (new task) and backward (old task) transfer degradation. By selectively unlearning interfering information before learning new data, our method prevents destructive interference and facilitates smoother optimization trajectories.
>
> It is important to note that our objective is not to directly measure generalization, but to regulate training dynamics that underlie catastrophic forgetting and negative transfer. In this sense, our gradient-conflict detection serves as a training-phase proxy for harmful transfer interactions, enabling proactive mitigation during optimization. Over the long term, this results in better retention, smoother task transitions, and higher overall test accuracy.
>
> ---
>
> > W2: The implementation of the transferability bound also has gaps. The use of LEEP scores as a proxy for λ isn't well justified, especially since LEEP was designed for offline transfer learning with fixed source models, not continual learning where models evolve. The connection between Eq.4 and the actual bound is hand-wavy at best.
>
> **Response:**
>
> The transferability bound in Eq. (1) (of main manuscript) includes a term $\lambda$ that reflects the label-space compatibility between source and target domain, and direct computation of $\lambda$ is intractable. Therefore, we approximate it using the LEEP score, which provides an empirical measure of transferability between tasks.
>
> Although LEEP was originally proposed for offline transfer learning, Nguyen et al. [5] explicitly highlight in their Applications of LEEP section that:
>
> > “Aside from transfer and meta-transfer learning, our LEEP scores are potentially useful for continual learning, multi-task learning, and feature selection. For instance, LEEP scores can be used to estimate the hardness of task sequences, thereby helping to analyze properties of continual learning algorithms.”
> >
>
> This provides direct theoretical support for our adaptation of LEEP to the continual learning setting. In our framework, before learning new knowledge, we treat the current model (which encapsulates all previously learned tasks) as the source mode $f_ {\theta_ S}$, and the incoming task data as the target data $X_T$. The computed LEEP score thus directly evaluates the compatibility between old and new knowledge, i.e., how well the accumulated source representations explain the target distribution. A higher LEEP indicates stronger alignment and lower risk of negative transfer, while a lower score signals potential interference. This localized use maintains the fixed source assumption of LEEP while allowing online estimation of transferability as the model evolves.
>
> Therefore, our use of LEEP as a proxy for $\lambda$ is both theoretically grounded and practically motivated, it captures real-time knowledge compatibility, aligns with transferability bound, and is consistent with the LEEP authors’ own suggestion of its applicability to continual learning.

---

> > ### Author Response · Authors · 2025-11-22
> > **Official Comment by Authors**
> >
> > ---
> >
> > > W3 & Q2: The paper doesn't clearly explain what constitutes **interfering knowledge** versus **useful knowledge**. How exactly does maximizing cross-entropy loss lead to selective unlearning of only interfering knowledge? What prevents this from removing useful knowledge as well?
> >
> > **Response:**
> >
> > In continual learning, interfering knowledge refers to previously acquired information whose parameter gradients conflict with those of the new task, thereby hindering its learning and inducing negative transfer [1]. In contrast, useful knowledge denotes information encoded in parameters that are highly sensitive and important to previous tasks, i.e., parameters with high Fisher Information (FIM) values [6].
> >
> > From a theoretical perspective, the unlearning process minimizes the KL divergence $\mathrm{KL}(\rho_t \| \rho_u)$ between the current CL model parameter posterior $\rho_t$ and the target unlearned model parameter posterior $\rho_u$ . Following [7], we define the target unlearned posterior as an energy function:
> >
> > $\rho_u = e^{-\omega} \quad \text{and} \quad \omega = -L_{CL}.$
> >
> > This KL divergence can be further decomposed as:
> >
> > $\mathrm{KL}(\rho_t \| \rho_u) = \int \rho_t(\theta) \log \frac{\rho_t(\theta)}{\rho_u(\theta)} d\theta
> > = -\int \rho_t(\theta)\log\rho_u(\theta)d\theta + \int \rho_t(\theta)\log\rho_t(\theta)d\theta
> > = -E_{\rho_t}\log\rho_u + E_{\rho_t}\log\rho_t
> > = -E_{\rho_t}L_{CL} + E_{\rho_t}\log\rho_t$
> >
> > $\rho = \arg\min_\rho \mathrm{KL}(\rho_t \| \rho_u)
> > = \arg\min_\rho \mathcal{E}(\rho)
> > = -E_{\rho_t}L_{CL} + E_\rho \log\rho$
> >
> > where the second equation is to unlearn on the current mini-batch by optimizing an energy functional in function space over the CL parameter posterior distributions. Given that the energy functional $\mathcal{E}(\rho)$, as defined in the second equation, represents the negative loss of $L_{CL}$, it effectively promotes an increase in loss.
> >
> > To ensure selective unlearning, our LUM is explicitly regularized using the Fisher Information Matrix (Eq. 9). Therefore, the unlearning update focuses precisely on low-importance, interfering prior knowledge that would cause negative transfer during new knowledge acquisition, while high-FIM parameters that encode useful and task-critical information are protected by the unlearning step.
> >
> > Importantly, this mechanism is not coarse-grained: the combination of gradient-conflict detection and FIM-based regularization makes the unlearning process selective and structured, rather than broadly degrading performance. In practice, we observe that this selective unlearning improves both plasticity (learning new knowledge) and stability (preserving old knowledge).
> >
> > ---
> >
> > > W4: The global unlearning module appears disconnected from negative transfer detection, as it periodically resets neurons regardless of whether negative transfer is detected. This seems more like a generic plasticity mechanism than targeted unlearning.
> >
> > **Response:**
> >
> > Although GUM is always active and does not depend on negative transfer detection, it is not a generic plasticity mechanism. Instead, GUM performs targeted unlearning exclusively on neurons that are simultaneously low-activity and low-importance, ensuring that only uninformative and outdated neurons are reset. This allows the model to recover useful capacity without disrupting essential knowledge, thereby enhancing long-term plasticity in a principled and controlled manner.
> >
> > In contrast, the LUM is detection-driven. It is activated only when negative transfer is detected and performs local unlearning to remove interfering knowledge before learning new mini-batch data. Thus, GUM and LUM serve complementary purposes:
> >
> > - LUM addresses immediate interference and improves short-term adaptation.
> > - GUM maintains long-term adaptability and prevents representational saturation by selectively freeing unused capacity.
> >
> > Their synergistic effect is clearly demonstrated in Appendix A.3.5 (ablation study), where combining LUM and GUM yields further reductions in negative backward transfer and achieves the best overall performance compared to using either module alone.
> >
> > In summary, GUM is not a generic plasticity mechanism but a targeted, FIM-guided unlearning process that works in tandem with LUM to maintain a dynamic balance between stability and plasticity throughout continual learning.

---

> > > ### Author Response · Authors · 2025-11-22
> > > **Official Comment by Authors**
> > >
> > > ---
> > >
> > > >W5: The paper claims that unlearning improves plasticity but provides no direct evidence for this. While they show accuracy improvements on new tasks, they don't demonstrate that this is due to increased plasticity rather than other factors. Including analysis of network capacity in [1] could help.
> > >
> > > **Response:**
> > >
> > > In the referenced paper, plasticity is defined explicitly as the model’s ability to acquire new knowledge, and the loss of plasticity refers to the phenomenon whereby deep models gradually lose their ability to learn new data as training progresses. This aligns exactly with our operational definition in the paper.
> > >
> > > As stated in the manuscript, we assess plasticity using the accuracy on newly introduced tasks, since this directly reflects the model’s ability to effectively learn novel information. Under this definition, the consistent improvements we report on new-task accuracy across benchmarks provide clear empirical evidence that our method enhances plasticity.
> > >
> > > Importantly, the improved plasticity is not solely due to reclaimed capacity from GUM. LUM also mitigates negative transfer and reduces harmful interference, enabling more efficient learning of new tasks.
> > >
> > > To further strengthen the characterization of plasticity, we analyze network capacity following the neuron-level activation perspective in [1]. According to their findings, plasticity deteriorates when many neuron-like units become dormant, meaning that a large portion of the network stops responding to new data and thus cannot support further learning. We measure the number of such weak units, neurons whose average activation falls below a threshold (e.g., <0.3), and observe that our method effectively reduces the number of weak units in later tasks compared to the baseline, as shown in Table R1. This indicates that our method effectively reclaims low-contributing neurons, enabling the network to support long-term learning.
> > >
> > > Table R1: The number of weak units (CIFAR-100).
> > >
> > > | **Method** | Task 1 | Task 2 | Task 3 | Task 4 | Task 5 | Task 6 | Task 7 | Task 8 | Task 9 | Task 10 |
> > > | --- | --- | --- | --- | --- | --- | --- | --- | --- | --- | --- |
> > > | DER++ | **5** | **8** | 7 | **12** | **15** | **22** | **28** | **35** | **40** | **42** |
> > > | w/ OUR(G) | 4 | 7 | **9** | 8 | 10 | 18 | 11 | 13 | 29 | 11 |
> > >
> > > ---
> > >
> > > > Q3: In Tab. 13, the LUM activation frequency varies dramatically across datasets (92% for CIFAR-100 epoch-level vs 36% for batch-level). What does this suggest about the reliability of the detection mechanism?
> > >
> > > **Response:**
> > >
> > > The variation in LUM activation frequency across datasets and detection granularities should not be interpreted as unreliability. Instead, it reflects the inherent differences between coarse-grained and fine-grained detection.
> > >
> > > Epoch-level detection aggregates gradients over an entire epoch, producing a stronger and more sensitive signal that naturally results in more frequent activations. In contrast, batch-level detection examines conflicts at mini-batch granularity and is therefore more selective and precise, activating the LUM only when genuine interference occurs, hence its much lower activation rate. Crucially, both detection granularities consistently improve performance across datasets (in Table 14 of Appendix A.4.2), confirming that gradient conflict is a reliable indicator of negative transfer.
> > >
> > > The fact that batch-level detection achieves higher accuracy with fewer activations further demonstrates that the mechanism is highly targeted. Overall, the difference in activation frequency reflects sensitivity vs. selectivity, not unreliability, and the consistent accuracy gains validate the robustness of our detection strategy.

---

> > > > ### Author Response · Authors · 2025-11-22
> > > > **Official Comment by Authors**
> > > >
> > > > ---
> > > >
> > > > > Q4: Could the performance improvements be explained by regularization effects rather than actually addressing negative transfer or improving plasticity?
> > > >
> > > > **Response:**
> > > >
> > > > While our framework includes a regularization term during the learning stage, we emphasize that this term is not the main driver of the observed performance gains. To validate this, we conducted an additional ablation study in which we completely removed this regularization item while keeping all other components (detection, LUM, and GUM) unchanged. The results in Table R2 show that the majority of improvements persist, confirming that the improvements stem from our unlearning mechanism rather than from generic regularization effects.
> > > >
> > > > Moreover, we provide detailed analyses of both new-task learning gains and old-task retention gains in the Figures 3-4 (of main manuscript). These results demonstrate that our framework simultaneously improves plasticity (better acquisition of new tasks) and stability (better preservation of early tasks), which cannot be explained by a simple regularizer.
> > > >
> > > > The key contributions arise from the complementary roles of our two unlearning modules:
> > > >
> > > > - LUM performs selective batch-level unlearning, removing interfering prior knowledge that would otherwise cause negative transfer;
> > > > - GUM restores long-term plasticity by reclaiming low-contributing neurons, expanding the network’s usable capacity over time.
> > > >
> > > > Together, these components, validated through ablation, form the core mechanism that addresses negative transfer and enhances continual learning performance. The regularization term provides only minor auxiliary benefit, whereas the substantial improvements are attributable to explicitly detecting and unlearning interfering prior knowledge.
> > > >
> > > > Table R2: Ablation study of the our proposed framework (DEDUCE).
> > > >
> > > > | **Method** | **CIFAR-100 CIL** | **CIFAR-100 TIL** | **CIFAR-10 CIL** | **CIFAR-10 TIL** | **Tiny-ImageNet CIL** | **Tiny-ImageNet TIL** | **CORE-50 CIL** | **CORE-50 TIL** |
> > > > | --- | --- | --- | --- | --- | --- | --- | --- | --- |
> > > > | DER++ | 36.8±0.9 | 75.6±0.5 | 71.1±1.3 | 93.4±0.2 | 15.6±1.9 | 51.1±0.7 | 40.1±1.5 | 76.0±2.0 |
> > > > | w/ LUM | 39.4±0.6 | 76.6±0.3 | 72.6±0.6 | 93.9±0.5 | 16.6±0.5 | 52.6±0.9 | 40.5±1.4 | 76.3±1.2 |
> > > > | w/ GUM | 37.7±0.3 | 76.4±0.5 | 72.4±0.1 | 93.6±0.0 | 16.3±0.8 | 52.2±0.9 | **42.1±1.2** | **77.4±1.4** |
> > > > | wo/ Regularization | 39.6±0.3 | 77.2±0.2 | 73.6±0.3 | 94.0±0.3 | 19.6±0.7 | 53.8±0.6 | 41.0±1.1 | 76.5±0.8 |
> > > > | w/ OUR(G) | **39.8±0.9** | **78.4±0.4** | **74.6±0.7** | **94.8±0.2** | **22.1±0.5** | **55.6±0.1** | 41.2±1.4 | 76.8±1.1 |
> > > >
> > > > ---
> > > >
> > > > References:
> > > >
> > > > [1] Learning to learn without forgetting by maximizing transfer and minimizing interference. ICLR, 2019.
> > > >
> > > > [2] A comprehensive survey of continual learning: Theory, method and application. 2024, TPAMI.
> > > >
> > > > [3] Afec: Active forgetting of negative transfer in continual learning. NeurIPS, 2021.
> > > >
> > > > [4] Continual learning with global alignment. NeurIPS, 2024.
> > > >
> > > > [5] Leep: a new measure to evaluate transferability of learned representations. ICML, 2020.
> > > >
> > > > [6] Overcoming catastrophic forgetting in neural networks. PNAS, 2017.
> > > >
> > > > [7] Sampling as optimization in the space of measures: The langevin dynamics as a composite optimization problem. PMLR, 2018.

---

### Official Review · Reviewer_6M6c · 2025-10-29

**Soundness:** 3
**Presentation:** 3
**Contribution:** 3
**Rating:** 4
**Confidence:** 5

**Summary:**

This paper addresses the problem of negative transfer in continual learning (CL) by proposing a 3-stage framework, Detect, Decide, and Unlearn, which can be plugged into existing CL methods to improve the performance. It presented two strategies, task-level transferability bound and batch-level gradient conflict analysis, to detect negative transfer. If the interference between old and new tasks is detected, it performs unlearn-learn update to future batches. This approach was tested on several existing CL approaches with standard CL benchmarks, showing its effectiveness of increasing the CL performance.

**Strengths:**

The paper has a novel contribution to CL, i.e., focusing on negative transfer, while many others are concerned with catastrophic forgetting.

The two different mechanisms of detecting negative transfers are designed at different levels and can complement each other.

The LUM is designed to balance between removing the interference (negative transfer) and important parameters (prior knowledge). The GUM can help keep those low activated but important neurons.

**Weaknesses:**

This detect-decide-unlearn process introduces significant computational costs, and the partial of the costs is mentioned in the appendix. The work needs to be further analyze the extra costs and be optimized to make this the process lightweight.

In Eq8, it is unclear why L_{unlearn} is defined in that way. Maximizing the error on the new task's current batch does not necessary mean to unlearn the old task. It could be unlearning the new task as well.

The LUM relies on a diagonal FIM to compute the importance of parameters, which makes it inherits all the known limitations of diagonal FIM, such as inaccurate approximation,  scalability issue of storing FIM, and saturation of FIM with more tasks are coming.

The experiment of the paper should include several more recent CL baselines, including those leveraging pre-trained models, to give a more comprehensive evaluation.

**Questions:**

1. Clarify Eq8, i.e., how does maximizing the loss on the current batch can reliably unlearn prior interfering knowledge, rather than unlearning new knowledge
2. Provide a more detailed analysis on the extra computational cost and propose/discuss optimiziation to make the framework more lightweight
3. How to address the limitations of using diagonal FIM?
4. How will the proposed framework perform with more recent CL including those using pre-trained models?

---

> ### Author Response · Authors · 2025-11-22
> **Official Comment by Authors**
>
> >W1 & Q2: The work needs to be further analyze the extra costs and be optimized to make this the process lightweight. Provide a more detailed analysis on the extra computational cost and propose/discuss optimiziation to make the framework more lightweight
>
> **Response:**
>
> We thank the reviewer for raising this point. To quantify the computational overhead introduced by our method, we report training time per epoch on the CIFAR-100 dataset for various baselines, with and without our proposed modules, as shown in Table R1:
>
> Table R1: Computation efficiency of the proposed method (CIFAR-100, one epoch in seconds).
>
> | Method | Baseline | w/ OUR(B) | w/ OUR(G) |
> | --- | --- | --- | --- |
> | PCR | 11.19s | — | 57.38s |
> | MOSE | 112.32s | — | 235.23s |
> | oEWC | 6.94s | — | 61.30s |
> | STAR | 23.12s | 37.46s | 85.06s |
> | AGEM | 10.99s | 29.17s | 74.77s |
> | HAL | 13.60s | 17.66s | 86.50s |
> | ER | 11.23s | 16.49s | 68.69s |
> | DER++ | 11.90s | 19.73s | 74.88s |
>
> As shown in Table R1, while our detect–decide–unlearn framework introduces additional computation, the overhead remains moderate. Importantly, the bound-based variant OUR(B) adds minimal extra cost, its runtime is very close to the original baselines (e.g., HAL: 13.60s → 17.66s; DER++: 11.90s → 19.73s). This provides a highly efficient option for settings with limited computational budgets.
>
> The gradient-based variant (OUR(G)) incurs higher overhead because it performs batch-wise detection. However, its runtime is still within a reasonable range, and several baselines combined with OUR(G) (e.g., DER++[1] + OUR(G), STAR [2] + OUR(G)) are still faster than MOSE [3] alone, indicating that the cost is far from prohibitive.
>
> Specifically, our method requires one extra forward pass for detection and one forward/backward pass for unlearning when unlearning is triggered. In many cases, unlearning is not activated, resulting in only a single additional forward pass.
>
> To further balance efficiency and performance, we provide multiple detection strategies with different granularities (task-level, epoch-level, and batch-level). These options act as built-in optimization strategies: users can choose the appropriate granularity depending on computational constraints, while all variants still provide consistent performance improvements (Appendix A.4.2).
>
> ---
>
> > W2 & Q1: In Eq8, it is unclear why $L_{unlearn}$ is defined in that way. Clarify Eq8, i.e., how does maximizing the loss on the current batch can reliably unlearn prior interfering knowledge, rather than unlearning new knowledge
>
> **Response:**
>
> Our definition of $L_{unlearn}$ follows the theoretical formulation in [4], where the goal of unlearning is to move the model parameters from the current posterior $\rho_t$ toward the target unlearned posterior $\rho_u$ . This is achieved by minimizing the KL divergence $\mathrm{KL}(\rho_t \| \rho_u)$, where  $\rho_u$  is defined as an energy-based posterior proportional to  $\rho_u = e^{-\omega} \quad \text{and} \quad \omega = -L_{CL}.$
>
> This KL divergence can be further decomposed as:
>
> $\mathrm{KL}(\rho_t \| \rho_u) = \int \rho_t(\theta) \log \frac{\rho_t(\theta)}{\rho_u(\theta)} d\theta
> = -\int \rho_t(\theta)\log\rho_u(\theta)d\theta + \int \rho_t(\theta)\log\rho_t(\theta)d\theta
> = -E_{\rho_t}\log\rho_u + E_{\rho_t}\log\rho_t
> = -E_{\rho_t}L_{CL} + E_{\rho_t}\log\rho_t$
>
> $\rho = \arg\min_\rho \mathrm{KL}(\rho_t \| \rho_u)
> = \arg\min_\rho \mathcal{E}(\rho)
> = -E_{\rho_t}L_{CL} + E_\rho \log\rho$
>
> where the second equation is to unlearn on the current mini-batch by optimizing an energy functional in function space over the CL parameter posterior distributions. Given that the energy functional $\mathcal{E}(\rho)$, as defined in the second equation, represents the negative loss of $L_{CL}$, it effectively promotes an increase in loss.
>
> To ensure selective unlearning, our LUM is explicitly regularized using the Fisher Information Matrix (Eq. 9). Therefore, the unlearning update focuses precisely on low-importance, interfering prior knowledge that would cause negative transfer during new knowledge acquisition.
>
> Importantly, because the unlearning step precedes the learning of new batches, it cannot unlearn the new knowledge. Instead, it selectively unlearns interfering prior knowledge, which may negatively transfer to the current batch, thereby preparing the model to effectively acquire new knowledge.

---

> > ### Author Response · Authors · 2025-11-22
> > **Official Comment by Authors**
> >
> > ---
> >
> > > W3 & Q3: The LUM relies on a diagonal FIM to compute the importance of parameters, which makes it inherits all the known limitations of diagonal FIM. How to address the limitations of using diagonal FIM?
> >
> > **Response:**
> >
> > We acknowledge that using a full Fisher Information Matrix (FIM) is theoretically more accurate. However, computing and storing the full FIM is infeasible for modern neural networks. To illustrate the computational burden, consider a ResNet-18 architecture, which contains approximately ≈ 11 million trainable parameters. The full FIM would therefore be a matrix of size: $11M \times 11M \approx 1.21 \times 10^{14} \text{ elements}$. Storing this matrix in 32-bit floating-point numbers requires: $1.21\times10^{14} \text{ elements} \times 4\text{ bytes}
> > \approx 484 \text{ terabytes}$.
> >
> > Therefore, calculating and storing the full FIM is practically infeasible, and this is precisely why almost all continual learning methods [5]-[8] rely on a diagonal approximation. Our method builds upon this widely accepted approximation, ensuring computational efficiency and compatibility with existing CL pipelines.
> >
> > Our method adopts the diagonal FIM approximation for scalability, but it differs from the vanilla EWC formulation by introducing a predictive-distribution–weighted estimation. Concretely, instead of directly accumulating squared gradients as in EWC, we weight each term by the model’s belief $p(y|x)$, using the $F_i \leftarrow p(y|x) \cdot \left(\frac{\partial \log p(y|x)}{\partial\theta_i}\right)^2$.  This mechanism better reflects the expected Fisher term [9]: $F _\theta
> > = \mathbb{E} _{z}
> > \left[
> > (\nabla _\theta \log p _\theta(z))^\top
> > (\nabla _\theta \log p _\theta(z))
> > \right], z=(x,y),$
> >  suppresses noisy gradients from uncertain samples, and makes the diagonal Fisher a closer approximation to the true Fisher expectation while preserving computational efficiency.
> >
> > In the future, we would like to explore further optimization schemes for FIM by considering Kronecker-factored curvature (K-FAC) [9] or low-rank blockwise approximation [10], though this falls outside the scope of this paper's core research.
> >
> > ---
> >
> > >W4 & Q4: The experiment of the paper should include several more recent CL baselines, including those leveraging pre-trained models, to give a more comprehensive evaluation. How will the proposed framework perform with more recent CL including those using pre-trained models?
> >
> > **Response:**
> >
> > Pre-trained-based CL typically focuses on rapid adaptation to downstream tasks with a strong prior representation, often by fine-tuning a frozen or partially frozen backbone. In such scenarios, the backbone parameters are largely fixed, leaving very limited opportunities for unlearning, since most parameters cannot be modified.
> >
> > In contrast, our work targets the general continual learning problem, where the model must continuously balance stability and plasticity while learning from scratch or from randomly initialized networks. In this setting, negative transfer arises from interference between evolving feature representations, precisely the situation that our detect–decide–unlearn framework is designed to address. We also note that further work is needed to adapt DEDUCE to continual pre-training scenarios, where the backbone evolves over time and thus offers more opportunities for unlearning to take effect.
> >
> > ---
> >
> > References:
> >
> > [1] Dark experience for general continual learning: a strong, simple baseline. NeurIPS, 2020.
> >
> > [2] STAR: Stability-inducing weight perturbation for continual learning. ICLR, 2025.
> >
> > [3] Orchestrate latent expertise: Advancing online continual learning with multi-level supervision and reverse self-distillation. CVPR, 2024.
> >
> > [4] Sampling as optimization in the space of measures: The langevin dynamics as a composite optimization problem. PMLR, 2018.
> >
> > [5] Overcoming catastrophic forgetting in neural networks. PNAS, 2017.
> >
> > [6] Continual learning through synaptic intelligence. ICML, 2017.
> >
> > [7] Memory Aware Synapses: Learning what (not) to forget. ECCV, 2018.
> >
> > [8] On the computation of the Fisher Information in continual learning. ICLR blogpost track, 2025.
> >
> > [9] Optimizing neural networks with Kronecker-factored approximate curvature. ICML, 2015.
> >
> > [10] Revisiting natural gradient for deep networks. ICLR, 2014.

---

### Official Review · Reviewer_aygx · 2025-10-31

**Soundness:** 3
**Presentation:** 3
**Contribution:** 3
**Rating:** 4
**Confidence:** 3

**Summary:**

This paper proposes a novel approach to continual learning that aims to effectively balance stability and plasticity. The key idea is to detect interference from new tasks on past task knowledge. When such interference is detected, a Local Unlearning Module selectively removes conflicting knowledge while preserving critical parameters. A Global Unlearning Module identifies less important network components to allocate more learning capacity for new tasks, thereby reducing forgetting. The method is supported by theoretical justifications and evaluated empirically on standard CL benchmarks.

**Strengths:**

- A novel and well-motivated approach that integrates neuron importance estimation with targeted unlearning based on interference signals.

- Strong theoretical foundation supporting the role of local and global unlearning.

- The proposed method indicates a meaningful contribution to continual learning research.

**Weaknesses:**

The evaluation setup requires more clarity. See details below:
- It is unclear whether all methods (including the proposed one) are trained in a fully online continual learning scenario.

- The buffer size used in Table 1 should be explicitly stated.

- The DER++ baseline results differ noticeably from previously published online CL work (e.g., OnPro). I suggest re-running DER++ and comparing under the OnPro (ICCV 2023) experimental protocol for fairness:

  - OnPro: Online Prototype Learning for Online Continual Learning, ICCV 2023

- Results for the STAR baseline also differ from what has been reported in prior work. How is STAR implemented here?
  - According to the STAR paper, STAR > DER++ in most settings, so STAR being significantly weaker raises concerns of evaluation fairness.
  - When comparing with STAR, shouldn’t the comparison be against X-DER + STAR?

**Questions:**

- What is the explanation for the results where the bound-based variant (OUR(B)) underperforms the gradient-based variant (OUR(G))?

- Why does backward transfer (Figure 4) decrease for HAL when combined with the proposed method?

- Why does EWC perform worse than the simple fine-tuning baseline in your experiments?

Also refer to the weakness section above.


**Additional Remarks:**

I am open to increasing my score if the concerns about evaluation fairness and clarity are addressed in the rebuttal.

---

> ### Author Response · Authors · 2025-11-22
> **Official Comment by Authors**
>
> > W1: It is unclear whether all methods (including the proposed one) are trained in a fully online continual learning scenario.
>
> **Response:**
>
> We appreciate the reviewer’s attention to the evaluation setup. Our proposed framework is fully compatible with the online continual learning setting, and whenever we compare against online CL baselines such as PCR [1] and MOSE [2], we train strictly under the fully online scenario. For comparisons with rehearsal-based or regularization-based methods, we adopt their original training protocols to ensure fairness and reproduce their standard results.
>
> All training configurations, including the number of epochs, batch size, and other implementation details, are clearly described in Appendix A.3.4. We will make this correspondence more explicit in the revised version.
>
> ---
>
> > W2: The buffer size used in Table 1 should be explicitly stated.
>
> **Response:**
>
> The buffer size used in Table 1 is 500, which is consistent across all methods for a fair comparison (see Appendix A.3.4 for details). We will explicitly state this in the main text to enhance clarity.
>
> ---
>
> >W3: The DER++ baseline results differ noticeably from previously published online CL work (e.g., OnPro). I suggest re-running DER++ and comparing under the OnPro (ICCV 2023) experimental protocol for fairness.
>
> **Response:**
>
> The DER++ baseline in our work was implemented strictly following the original paper’s setting, and our reproduced results are consistent with those reported in the original DER++ publication [3]. This ensures that the observed performance differences are solely due to the introduction of our proposed DEDUCE module.
>
> In response to the reviewer’s request, we additionally re-ran DER++ under the OnPro (ICCV 2023) experimental protocol and report the results in Table R1 below. As shown, integrating our method still yields consistent improvements across benchmarks:
>
> Table R1: Performance of DER++ under the OnPro protocol.
>
> | **Method** | **CIFAR-100 CIL** | **CIFAR-100 TIL** | **CIFAR-10 CIL** | **CIFAR-10 TIL** | **Tiny-ImageNet CIL** | **Tiny-ImageNet TIL** | **CORE-50 CIL** | **CORE-50 TIL** |
> | --- | --- | --- | --- | --- | --- | --- | --- | --- |
> | DER++ | 12.4±0.4 | 56.6±0.3 | 44.8±1.6 | 82.0±0.2 | 6.9±0.6 | 37.7±0.4 | 28.2±1.0 | 66.5±0.9 |
> | w/ OUR(G) | **15.1±0.2** | **57.3±0.3** | **50.5±0.3** | **86.8±0.6** | **8.8±0.4** | **41.9±0.4** | **29.8±0.8** | **70.7±0.9** |
>
> These results confirm that our framework provides clear benefits even under the OnPro setting. At the same time, we respectfully note that OnPro’s protocol differs from the original DER++ training scheme. For clarity and fairness, we believe that comparing each baseline within its original training setting remains the most appropriate way to evaluate the incremental improvement contributed by DEDUCE. Nevertheless, we fully acknowledge the reviewer’s perspective, and the additional OnPro experiments reinforce the robustness of our method across different online CL protocols. We will include these results and clarifications in the revised version.
>
> ---
>
> >W4: Results for the STAR baseline also differ from what has been reported in prior work. How is STAR implemented here?
>
> **Response:**
>
> The STAR [4] results reported in Table 1 (of main manuscript) correspond to the DER++ + STAR configuration, which was implemented using the official STAR source code and original hyperparameter settings.
>
> Upon double-checking, we identified that in the STAR codebase, a scheduler was defined for the CIFAR-100 dataset, while the Continual-Dataset library we adopted also defines a scheduler by default. This caused an unintended scheduler conflict in our initial run, leading to STAR failing to demonstrate performance improvements over DER++ on the CIFAR-100. We have since corrected this conflict and re-run the experiments on CIFAR-100 following the original STAR hyperparameter setting. Meanwhile, we also add the comparisons with X-DER-RPC + STAR, as shown in Table R2.
>
> Table R2: Performance of baselines under the STAR experimental protocol.
>
> | **Method** | **CIFAR-100 CIL** | **CIFAR-100 TIL** | **CIFAR-10 CIL** | **CIFAR-10 TIL** | **Tiny-ImageNet CIL** | **Tiny-ImageNet TIL** | **CORE-50 CIL** | **CORE-50 TIL** |
> | --- | --- | --- | --- | --- | --- | --- | --- | --- |
> | DER++ | 36.8±0.9 | 75.6±0.5 | 71.1±1.3 | 93.4±0.2 | 15.6±1.9 | 51.1±0.7 | 40.1±1.5 | 76.0±2.0 |
> | DER+++STAR | 38.8±0.7 | 76.4±0.3 | 73.1±0.8 | 94.2±0.9 | 16.7±0.9 | 53.3±0.8 | 36.8±0.7 | 73.3±0.3 |
> | w/ OUR(G) | **40.9±0.8** | **77.7±0.3** | **74.7±0.6** | **95.5±0.5** | **18.5±0.6** | **55.1±0.8** | **42.2±0.2** | **78.5±0.5** |
> | X-DER(RPC) | 40.1±0.3 | 84.7±0.4 | 61.5±1.6 | 95.02±0.5 | 18.8±0.7 | 54.7±0.6 | 36.7±1.2 | 74.6±1.3 |
> | X-DER(RPC)+STAR | 41.4±0.6 | 84.8±0.3 | **63.8±1.6** | **95.4±0.3** | 19.7±0.4 | 55.7±0.3 | 38.4±0.6 | 75.9±0.4 |
> | w/ OUR(G) | **44.2±0.4** | **85.5±0.4** | 62.6±1.7 | 95.3±0.6 | **20.1±0.6** | **56.0±0.8** | **39.8±0.7** | **76.6±0.5** |

---

> > ### Comment · Reviewer_aygx · 2025-11-25
> >
> > Thank you for the clarifications. One thing I noticed is only a minor improvement from using your method when compared to already stronger baselines. Is it possible to compare your method with the OnPro baseline as well?

---

> ### Author Response · Authors · 2025-11-22
> **Official Comment by Authors**
>
> ---
>
> > Q1: What is the explanation for the results where the bound-based variant (OUR(B)) underperforms the gradient-based variant (OUR(G))?
>
> **Response:**
>
> The bound-based variant (OUR(B)) utilizes the transferability bound to detect potential negative transfer at the task level. Specifically, it performs detection after the first training epoch of a new task, and the result determines whether to activate the unlearning mechanism in subsequent epochs.
>
> In contrast, the gradient-based variant (OUR(G)) performs detection at the batch level, enabling much finer-grained identification of task interference. For each incoming batch, OUR(G) dynamically decides whether unlearning should be activated, adjusting the learning strategy in real time based on the current optimization state. This adaptive, batch-wise control allows the model to respond immediately to emerging conflicts, which explains why OUR(G) achieves higher performance.
>
> Nevertheless, it is important to note that both variants consistently outperform the baselines, demonstrating that our proposed transfer-aware unlearning mechanism is robust and beneficial across different detection granularities.
>
> ---
>
> > Q2: Why does backward transfer (Figure 4) decrease for HAL when combined with the proposed method?
>
> **Response:**
>
> As shown in Figures 3 and 5 (of main manuscript), integrating our method with HAL consistently improves both new-task learning and retention of early tasks on CIFAR-100 and TinyImageNet.
>
> The backward transfer decrease results from the definition of BWT [5]. BWT measures the difference between a task’s accuracy immediately after its training and its final accuracy after all tasks. Because our method substantially enhances HAL’s learning of new tasks, this initial accuracy becomes much higher, increasing the BWT gap even when both the initial and final accuracies improve.
>
> For example, for Task 2 in CIFAR-100 (Table R3), HAL’s accuracy drops from 52.2→2.9 (Δ=49.3), while HAL+OUR(G) drops from 83.6→13.2 (Δ=70.4). Although both 83.6 and 13.2 exceed the HAL baseline, the larger initial gain (83.6 vs. 52.2) yields a numerically larger drop, and thus a lower BWT.
>
> Table R3: Performance of Task 2.
>
> | **Method** | Task 1 | Task 2 | Task 3 | Task 4 | Task 5 | Task 6 | Task 7 | Task 8 | Task 9 | Task 10 |
> | --- | --- | --- | --- | --- | --- | --- | --- | --- | --- | --- |
> | HAL | — | 52.2 | 15.4 | 10.5 | 3.7 | 7.2 | 6.6 | 4.0 | 2.1 | 2.9 |
> | w/ OUR(G) | — | 83.6 | 45.1 | 30.9 | 20.4 | 18.5 | 16.8 | 14.3 | 8.3 | 13.2 |
>
> This effect also relates to HAL’s inherent design limitations. HAL anchors past activations to preserve old knowledge, providing strong stability but placing much tighter constraints on the learning of new tasks than on the retention of old ones. In other words, HAL’s static anchoring restricts the model’s ability to flexibly acquire new representations far more than it prevents forgetting. When our transfer-aware unlearning mechanism is integrated, it alleviates these constraints, leading to significant improvement in learning new tasks and moderate improvement in retaining old ones.
>
> Therefore, the reduced BWT in HAL actually reflects a stronger improvement in plasticity than in stability. This is why, in addition to reporting overall accuracy, we also explicitly provide new-task learning performance and retention of early tasks results to give a more complete and objective characterization of the learning dynamics. We will include further analysis in Appendix A.3.5.

---

> ### Author Response · Authors · 2025-11-22
> **Official Comment by Authors**
>
> ---
>
> >Q3: Why does EWC perform worse than the simple fine-tuning baseline in your experiments?
>
> **Response:**
>
> This behavior is expected and is reported in prior work [3]: online EWC performs worse than simple fine-tuning, especially in online continual learning. The core issue is that the EWC regularizer adds a constraint that forces the model to stay close to old-task optima. When the new-task gradient strongly conflicts with this regularization direction, the optimizer is pulled in competing directions, leading to ineffective parameter updates. As a result, the model fails to learn new knowledge sufficiently while still not successfully retaining old knowledge. In contrast, fine-tuning does not suffer from such optimization interference, allowing it to learn new tasks more effectively despite forgetting. Therefore, the weaker performance of oEWC relative to fine-tuning is a known phenomenon rather than an implementation issue.
>
> As shown in Table 1 (of main manuscript), the results for oEWC [6], PCR [1], and MOSE [2] are obtained under the fully online continual learning setting, where each sample is seen only once. In contrast, the fine-tuning results originally reported in Table 1 correspond to the standard offline training scenario.
>
> To provide a fair reference point, we additionally ran a fine-tuning baseline under the same fully online scenario as the online continual learning methods. The results are presented in Table R4, offering a more accurate performance comparison for the online setting.
>
> Table R4: Comparisons under online continual learning scenario.
>
> | **Method** | **CIFAR-100 CIL** | **CIFAR-100 TIL** | **CIFAR-10 CIL** | **CIFAR-10 TIL** | **Tiny-ImageNet CIL** | **Tiny-ImageNet TIL** | **CORE-50 CIL** | **CORE-50 TIL** |
> | --- | --- | --- | --- | --- | --- | --- | --- | --- |
> | Fine-tuning | 4.2±0.2 | 26.4±0.7 | 15.0±0.5 | 60.8±1.0 | 3.7±0.2 | 18.8±0.8 | 6.9±0.9 | 34.7±0.9 |
> | oEWC | 4.8±0.2 | 21.4±1.1 | 15.8±1.4 | 57.3±1.3 | 4.0±0.4 | 13.6±1.7 | 7.8±0.4 | 34.3±1.7 |
> | w/ OUR(G) | **5.6±0.6** | **27.9±1.3** | **16.4±1.0** | **62.9±1.1** | **4.8±0.5** | **20.3±1.4** | **7.9±0.3** | **35.0±1.3** |
>
> ---
>
> References:
>
> [1]  Proxy-based contrastive replay for online class-incremental continual learning. CVPR, 2023
>
> [2] Orchestrate latent expertise: Advancing online continual learning with multi-level supervision and reverse self-distillation. CVPR, 2024.
>
> [3] Dark experience for general continual learning: a strong, simple baseline. NeurIPS, 2020.
>
> [4] STAR: Stability-inducing weight perturbation for continual learning. ICLR, 2025.
>
> [5] A comprehensive survey of continual learning: Theory, method and application. 2024, TPAMI.
>
> [6] Progress & compress: A scalable framework for continual learning. ICML, 2018.

---

> ### Author Response · Authors · 2025-11-26
> **Official Comment by Authors**
>
> > One thing I noticed is only a minor improvement from using your method when compared to already stronger baselines. Is it possible to compare your method with the OnPro baseline as well?
>
> Thank you for the insightful comment. We would like to clarify that DEDUCE is designed as a **plug-in framework**, so its improvements are naturally constrained by the strength and design of the underlying baselines. Nonetheless, because **negative transfer persists across all methods**, DEDUCE consistently improves performance regardless of the baseline used.
>
> We evaluate DEDUCE across **multiple scenarios** (online/offline), **multiple settings** (CIL/TIL), and **multiple datasets**, and consistently observe **stable gains**. As expected, the improvement magnitude varies across baselines and datasets. For instance, in **Table R2**, STAR yields different benefits depending on the baseline and dataset, yet DEDUCE still provides **additional gains of up to ~5.4%**, demonstrating its robustness and complementary nature.
>
> Beyond the numerical improvements, DEDUCE introduces a **new perspective** to continual learning: using **selective unlearning** to regulate both forward and backward transfer, rather than focusing solely on mitigating forgetting. This mechanism is orthogonal to existing strategies and enhances their effectiveness.
>
> Following the reviewer’s suggestion, we have additionally compared DEDUCE with **OnPro** (buffer size=500, batch_size=32, buffer_batch_size=32, other parameters are set according to the original paper), as shown in Table R5. The results further confirm the stability and effectiveness of our approach.
>
> Table R5. Comparison with OnPro.
>
> | **Method** | **CIFAR-100 CIL** | **CIFAR-10 CIL** | **Tiny-ImageNet CIL** | **CORE-50 CIL** |
> | --- | --- | --- | --- | --- |
> | OnPro | 19.47±0.84 | 66.86±1.24 | 6.90±0.18 | 32.09±1.44 |
> | w/ OUR(G) | **21.43±0.14** | **68.41±0.78** | **7.63±0.18** | **34.73±0.48** |

---

### Official Review · Reviewer_GPEL · 2025-11-01

**Soundness:** 2
**Presentation:** 3
**Contribution:** 3
**Rating:** 6
**Confidence:** 3

**Summary:**

The paper proposes DEDUCE, a transfer-aware framework for continual learning that dynamically detects and mitigates negative transfer, where outdated or irrelevant prior knowledge interferes with learning new tasks. The framework combines two detection strategies: a transferability bound derived from domain adaptation theory to estimate when prior knowledge harms generalization, and gradient conflict analysis to identify real-time interference between old and new task gradients. Experiments across CIFAR100, TinyImageNet, and CORE50 show consistent improvements over strong CL baselines.

**Strengths:**

1. Introduces a novel approach to detect and decide when to unlearn, bridging machine unlearning and continual learning communities.
2. Dual-strategy detection plus hybrid unlearning are well-motivated and mathematically backed.

**Weaknesses:**

1. Relies on stored exemplars and known task identities, and the applicability to online task-free CL remains unclear.
2. All experiments use vision classification benchmarks, which may limit the paper's generality.
3. The framework adds two auxiliary modules, which may cause additional compute complexity.

**Questions:**

What is the computational overhead of the proposed method compared to other methods?

---

> ### Author Response · Authors · 2025-11-22
> **Official Comment by Authors**
>
> >W1: Relies on stored exemplars and known task identities, and the applicability to online task-free CL remains unclear.
>
> **Response:**
>
> We thank the reviewer for the thoughtful comment. Our current implementation indeed relies on stored exemplars and known task identities. However, we also evaluate the scenario with a small buffer size (100 samples), demonstrating that even under highly constrained memory conditions, our method consistently improves the performance of existing continual learning baselines in both class-incremental and task-incremental settings (Tables 10-11 in Appendix A.3.5). Furthermore, our gradient conflict analysis–based detection mechanism is naturally compatible with online continual learning and does not require explicit task boundaries, making it promising for future extensions to the online task-free setting.
>
> The core innovation of DEDUCE lies in reframing continual learning beyond the traditional stability–plasticity trade-off. Instead of treating forgetting as inherently detrimental, we explicitly detect and selectively unlearn prior knowledge that induces negative transfer and task interference. This transfer-aware unlearning stands in contrast to existing methods, which typically rely on static regularization or heuristic constraints to preserve past knowledge.
>
> In summary, while our current work focuses on the exemplar-based, task-aware regime, DEDUCE advances continual learning by explicitly incorporating selective, transfer-aware unlearning, providing a principled and practical alternative to traditional stability–plasticity balancing. We will make these distinctions and potential extensions clearer in the revised version.
>
> ---
>
> > W2: All experiments use vision classification benchmarks, which may limit the paper's generality.
>
> **Response:**
>
> The primary research scope of our work is continual learning for vision classification, focusing on how to dynamically detect negative transfer and mitigate it through a hybrid unlearning mechanism. The vision benchmarks used in our experiments are standard and widely recognized in the continual learning community for evaluation.
>
> While our experiments are conducted in the vision domain, our framework is not restricted to any vision-specific assumptions. The proposed detect–decide–unlearn mechanism and transfer-aware unlearning formulation provide a general and principled perspective on continual learning. Here, we wish to clarify that negative transfer is not limited to any specific modality, and the underlying optimization conflicts that motivate our approach also appear in domains such as NLP, time-series forecasting, and reinforcement learning. Therefore, our method has clear potential for extension to other modalities, even though the present work focuses on vision tasks.
>
> ---
>
> >W3 & Q1: The framework adds two auxiliary modules, which may cause additional compute complexity. What is the computational overhead of the proposed method compared to other methods?
>
> **Response:**
>
> To quantify the computational overhead introduced by our method, we report training time per epoch on the CIFAR-100 dataset for various baselines, with and without our proposed modules, as shown in Table R1:
>
>
> Table R1. Computation efficiency of the proposed method (CIFAR-100, one epoch in seconds).
>
> | Method | Baseline | w/ OUR(B) | w/ OUR(G) |
> | --- | --- | --- | --- |
> | PCR | 11.19s | — | 57.38s |
> | MOSE | 112.32s | — | 235.23s |
> | oEWC | 6.94s | — | 61.30s |
> | STAR | 23.12s | 37.46s | 85.06s |
> | AGEM | 10.99s | 29.17s | 74.77s |
> | HAL | 13.60s | 17.66s | 86.50s |
> | ER | 11.23s | 16.49s | 68.69s |
> | DER++ | 11.90s | 19.73s | 74.88s |

---

> > ### Author Response · Authors · 2025-11-22
> > **Official Comment by Authors**
> >
> > As shown in Table R1, while our detect–decide–unlearn framework introduces additional computation, the overhead remains moderate. Importantly, the bound-based variant OUR(B) adds minimal extra cost, its runtime is very close to the original baselines (e.g., HAL: 13.60s → 17.66s; DER++: 11.90s → 19.73s). This provides a highly efficient option for settings with limited computational budgets.
> >
> > The gradient-based variant (OUR(G)) incurs higher overhead because it performs batch-wise detection. However, its runtime is still within a reasonable range, and several baselines combined with OUR(G) (e.g., DER++[1] + OUR(G), STAR [2] + OUR(G)) are still faster than MOSE [3] alone, indicating that the cost is far from prohibitive.
> >
> > Specifically, our method requires one extra forward pass for detection and one forward/backward pass for unlearning when unlearning is triggered. In many cases, unlearning is not activated, resulting in only a single additional forward pass.
> >
> > To balance efficiency and performance, we provide multiple detection strategies with different granularities (task-level, epoch-level, and batch-level). These options act as built-in optimization strategies: users can choose the appropriate granularity depending on computational constraints, while all variants still provide consistent performance improvements (Appendix A.4.2).
> >
> > ---
> >
> > References:
> >
> > [1] Dark experience for general continual learning: a strong, simple baseline. NeurIPS, 2020.
> >
> > [2] STAR: Stability-inducing weight perturbation for continual learning. ICLR, 2025.
> >
> > [3] Orchestrate latent expertise: Advancing online continual learning with multi-level supervision and reverse self-distillation. CVPR, 2024.

---

### Author Response · Authors · 2025-11-24
**Global Response**

We thank all reviewers for their thoughtful and constructive feedback. We have carefully revised the main manuscript to address all comments. Below we summarize the major changes made to improve clarity, theoretical grounding, and empirical completeness:

### **1. Expanded empirical evidence**

- We added extensive **computational cost analysis** (training time) and clarified that our multi-granularity detection strategies (batch-, epoch-, task-level) provide **built-in trade-offs** between accuracy and computation in **Appendix A.3.5 and A.4.2**. (Reviewer GPEL, 6M6c)
- We added new explanations and results in **Table 2 and Appendix A.3.5** showing that performance gains persist **even when removing the regularization term**, confirming that improvements stem from LUM/GUM rather than generic regularization. (Reviewer wJFi)
- We ran **online fine-tuning and DER++ under the OnPro protocol,** and also re-ran **STAR** under the original protocol. The updated results (including online DER++, DER++/X-DER-RPC + STAR + DEDUCE) have been added to **Table 1**, ensuring fair comparison. (Reviewer aygx)

### **2. Clarifying the detect–decide–unlearn framework**

- We clarified the definition of LUM, interfering knowledge, useful knowledge, and how LUM lead to selective unlearning based on a KL-divergence unlearning formulation in **Section 3.3.1**. (Reviewer 6M6c, wJFi)
- We clarified why the bound-based variant (OUR(B)) underperforms the gradient-based variant (OUR(G)) in **Section 4.1**. (Reviewer aygx)
- We clarified why the BWT decrease for HAL when integrated with DEDUCE in **Appendix A.3.5**. (Reviewer aygx)

### **3. Enhanced explanation of experimental settings**

- We clarified which methods are trained under fully online vs. their original settings, and  added the **explicit buffer size (500)** used in **Table 1**. (Reviewer aygx)

Detailed responses to each reviewer are provided below. All revisions in the main manuscript are marked in **blue**. We are happy to provide any further clarification or engage in further discussion.

---

### Meta-Review · Area_Chair_iDU2 · 2026-01-06

**Summary:**

This paper received four reviews, two with an initial rating of 6 (= marginally above the acceptance threshold), and two with an initial rating of 4 (= marginally below the acceptance threshold). The authors provided a rebuttal to each of the four reviews, but unfortunately, none of the reviewers responded in depth to this rebuttal before the premature end of the discussion period. (Although one of the reviewers did briefly respond with a follow-up question.)

There are two main concerns that prevent me from full-heartedly recommending acceptance of this paper.

The first concern regards computational costs. As pointed out by reviewers GPEL and 6M6c, the improved performance of the DEDUCE method comes at the expense of higher computational costs. In the initial version of the paper this issue was not really addressed, but in response to comments from reviewers the authors added an empirical comparison of training times in the Appendix. This analysis shows that DEDUCE indeed adds a substantial additional computational overhead: DEDUCE(G) approximately triples training times, while the more lightweight DEDUCE(D) still increases training times by approximately 50%. These are substantial additional computational costs. Firstly, it is important that this limitation is clearly discussed in the main text. Secondly, it is an open question whether the gain in performance that DEDUCE provides is worth the additional computational costs.

The second important concern is related to the first weakness raised by Reviewer wJFi. This reviewer points out that gradient conflict during optimization does not necessarily result in poor test-time transfer between tasks, and that therefore there seems to be a fundamental conceptual issue with the gradient conflict detection mechanism. I agree with this concern. Reviewer wJFi asks for evidence that gradient conflict during optimization actually correlates with negative transfer at test time, but the authors do not really provide this. The authors argue that the generalization outcome on past and future tasks is directly shaped by how gradients from different tasks interact during training, but this sidesteps and does not directly address the issue raised by the reviewer. In practice it might be possible to use gradient conflict during training as an trigger, but I think it is important to discuss that this variable does not directly measure negative transfer at convergence.

The above two concerns, and in particular the first one, limit the contribution of the paper.

However, I do agree with the reviewers that this paper has several strong aspects, such as the focus on negative transfer, highlighting that blindly preserving all knowledge can hurt performance and the comprehensive empirical evaluation in terms of performance.

All in all, I agree with the reviewers that this paper is borderline, and that the decision could go either way.

As I expect that, had there been a full discussion phase, three our of the four reviewers would have (marginally) supported acceptance (see below for details about why I expect this), I recommend this paper to be accepted. I thereby strongly urge the authors to clearly discuss the trade-off regarding computational costs in the main text of the camera-ready version of the paper, as well to discuss the conceptual issue with gradient conflict detection mechanism and to make sure all experiment details are described.

**Reviewer Concerns:**

As discussed in more detail above, I think the two main concerns that are still outstanding regard the computational costs and the conceptual issue with the gradient conflict detection mechanism. Please see above for a more detailed discussion of these.

**Reviewer Scores:**

Had there been a full discussion phase, I expect the final ratings would have been 4, 6, 6 and 6.

Reviewer GPEL and Reviewer 6M6c both raise the concern that while the proposed method improves performance, this comes at the expense of higher computational costs. As discussed above, I think the authors’ response to this concern is not fully satisfactory, and I therefore expect that reviewers GPEL and 6M6c would probably not have raised their rating had there been a full discussion phase.

Reviewer aygx mostly asks for clarifications and for empirical comparisons against additional baselines. I think the author rebuttal in response to this review is quite convincing. The requested clarifications seem to be provided, although some experimental details could still be added (e.g., how is the FIM computed? Could that perhaps contribute to the relative low performance of oEWC?). Most of the requested additional baselines are provided as well. There is some indication that the benefit of using DEDUCE reduces when it is compared with a stronger baseline, but I agree with the authors that this is something that is to be expected. I expect that, had there been a full discussion phase, Reviewer aygx would probably have increased their rating from 4 to 6.

Reviewer wJFi raises the concern that gradient conflict during optimization does not necessarily result in poor test-time transfer between tasks, and that therefore there seems to be a fundamental conceptual issue with the gradient conflict detection mechanism. The reviewer also asks for evidence that gradient conflict during optimization actually correlates with negative transfer at test time. As discuss above, I think that the authors’ response to this concern and question is not fully satisfactory. I therefore expect that reviewer wJFi would probably not have raised their rating had there been a full discussion phase.

---

### Decision · Program_Chairs · 2026-01-26

Accept (Poster)